# Cooperative STAT/NF-κB signaling regulates lymphoma metabolic reprogramming and aberrant *GOT2* expression

Maren Feist[1,2], Philipp Schwarzfischer[2,3], Paul Heinrich[3,4], Xueni Sun[3,4], Judith Kemper[1], Frederike von Bonin[1], Paula Perez-Rubio[4,5], Franziska Taruttis[2,5], Thorsten Rehberg[5], Katja Dettmer [3,4], Wolfram Gronwald[2,3,4], Jörg Reinders[3], Julia C. Engelmann[2,4,5,8], Jan Dudek[6], Wolfram Klapper[2,4,7], Lorenz Trümper[1,2,4], Rainer Spang[2,4,5], Peter J. Oefner[3] & Dieter Kube[1,2,4]

Knowledge of stromal factors that have a role in the transcriptional regulation of metabolic pathways aside from c-Myc is fundamental to improvements in lymphoma therapy. Using a *MYC*-inducible human B-cell line, we observed the cooperative activation of STAT3 and NF-κB by IL10 and CpG stimulation. We show that IL10 + CpG-mediated cell proliferation of MYC^low cells depends on glutaminolysis. By $^{13}$C- and $^{15}$N-tracing of glutamine metabolism and metabolite rescue experiments, we demonstrate that GOT2 provides aspartate and nucleotides to cells with activated or aberrant Jak/STAT and NF-κB signaling. A model of GOT2 transcriptional regulation is proposed, in which the cooperative phosphorylation of STAT3 and direct joint binding of STAT3 and p65/NF-κB to the proximal *GOT2* promoter are important. Furthermore, high aberrant *GOT2* expression is prognostic in diffuse large B-cell lymphoma underscoring the current findings and importance of stromal factors in lymphoma biology.

[1] Clinic of Haematology and Medical Oncology, University Medical Centre Göttingen, Lower Saxony 37075 Göttingen, Germany. [2] Network BMBF eBio MMML MYC-SYS, 37099 Göttingen / 93053 Regensburg, Germany. [3] Institute of Functional Genomics, University of Regensburg, Bavaria 93053 Regensburg, Germany. [4] Network BMBF eMed MMML-Demonstrators, 37099 Göttingen / 93053 Regensburg, Germany. [5] Statistical Bioinformatics, Institute of Functional Genomics, University of Regensburg, Bavaria 93053 Regensburg, Germany. [6] Institute of Biochemistry, University Medical Centre Göttingen, Lower Saxony 37075 Göttingen, Germany. [7] Department of Pathology, Hematopathology Section, UKSH Campus Kiel, 24105 Kiel, Germany. [8] Present address: NIOZ Royal Netherlands Institute for Sea Research and Utrecht University, 1790 AB Den Burg, The Netherlands. Correspondence and requests for materials should be addressed to D.K. (email: dieter.kube@med.uni-goettingen.de)

There is growing evidence for a cooperative role of stromal factors in the shift from quiescence to the proliferation and transformation of lymphocytes[1–3]. A fundamental question is how microenvironmental factors and the intracellular processes activated by them regulate metabolic reprogramming. Further, a deeper understanding of the differences between processes activated by either stromal or well-known single or complex genetic drivers such as the proto-oncogene c-MYC (MYC) is crucial to the development of novel targeted cancer therapies.

MYC has been shown to increase glutaminolysis and glycolysis to support tumor cell proliferation[4]. A central role of glutaminolysis in resting and proliferating B-cells was shown for the MYC-inducible human B-cell line P493-6, which serves as a model of MYC-driven lymphoma or normal B-cells[5]. In this model, MYC drives glutamine (Gln) catabolism via the aberrant expression of glutaminase (GLS). Upregulation of GLS increases the conversion of Gln to glutamate (Glu), which can be further metabolized to α-ketoglutarate (α-KG)[6–8]. The latter is a key metabolite involved in energy production via the tricarboxylic acid cycle (TCA) and it is also used as a biosynthetic precursor. Notably, α-KG may be synthesized from Glu by two different mechanisms: deamination by glutamate dehydrogenases (GLUD1/2) or transamination by aspartate transaminases

(GOT1/2) or alanine transaminases (GPT1/2). The latter also generate additional non-essential amino acids from Gln and are known to be regulated by MYC[9,10].

Recently, we have shown distinct changes in global gene expression (GE) induced by stromal stimuli that activate pattern recognition receptors and thereby mediate signaling in B-cells[11]. We further identified a combination of factors of the B-cell and lymphoma microenvironment that are capable of promoting proliferation in MYC-deprived P493-6 cells through signal transducer and activator of transcription (STAT) 3 and nuclear factor-κB (NF-κB) comparable to MYC overexpression[12]. Two questions emerging from these studies are whether cell proliferation in this model B-cell line is accompanied by metabolic reprogramming and whether such reprogramming is different or similar to that induced by MYC. Furthermore, it is important to understand how metabolic GE is regulated in the absence of aberrant MYC. This is necessary, as aberrant STAT and NF-κB signaling networks have been described among others in diffuse large B-cell lymphoma (DLBCL) and Hodgkin's lymphoma (HL) (reviewed by refs. [13,14]). A significant proportion of patients with DLBCL do not respond fully to the standard immunochemotherapy. This indicates a critical need for a better understanding of the specific pathways perturbed. Interestingly, a first metabolic

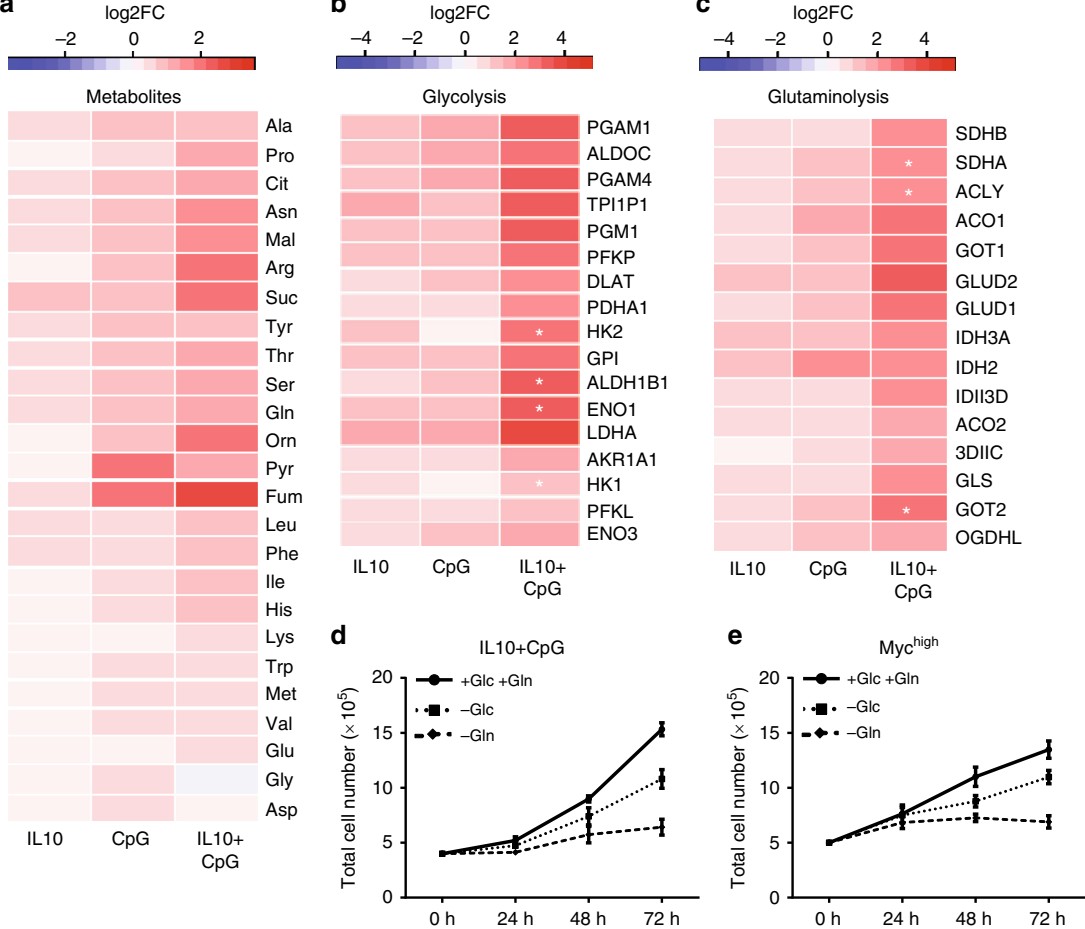

**Fig. 1** Activation of cell metabolism and global gene expression by combined stimulation of P493-6 cells with IL10 and CpG. **a** Heat map of changes in intracellular metabolite concentrations depicted as Log2FC after IL10 and/or CpG stimulation of P493-6 MYC[low] cells in relation to unstimulated cells. Heat map of gene expression changes associated with **b** glycolysis (KEGG-term) and **c** glutaminolysis (KEGG-term), respectively, after IL10 and/or CpG stimulation of P493-6 MYC[low] cells presented as Log2FC. Mean of three independent experiments is shown in all heat maps. Effects on gene expression were analyzed by linear regression and p-values for the IL10:CpG interaction terms were calculated (adjusted by Benjamini–Hochberg). Positive synergistic interactions are marked with a star (*$p < 0.05$). Relative cell counts of **d** IL10 + CpG-stimulated P493-6 MYC[low] cells and **e** unstimulated P493-6 MYC[high] cells grown in media with and without either glucose (–Glc) or glutamine (–Gln). Mean ± SD of three independent experiments are given

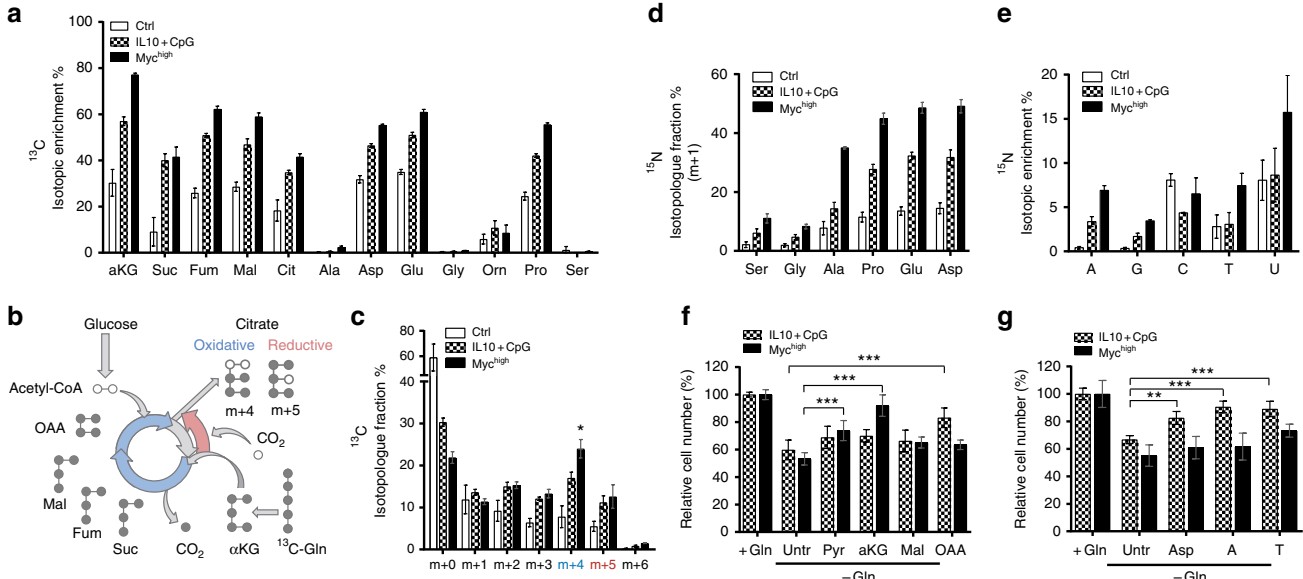

**Fig. 2** Proliferation of IL10 + CpG-stimulated and MYC-overexpressing cells is dependent on different glutamine-derived metabolites. **a** Mean isotopic enrichment of metabolites in unstimulated and IL10 + CpG-stimulated P493-6 MYC$^{low}$, as well as unstimulated P493-6 MYC$^{high}$ cells treated with $^{13}C_5$-glutamine within the text only 13C5-Gln is used?. **b** Scheme of oxidative decarboxylation (blue) and reductive carboxylation (red). Both can be distinguished by isotopic labelling in mass spectrometry. **c** Isotopic fractions of citrate in IL10 + CpG-stimulated P493-6 MYC$^{low}$ and P493-6 MYC$^{high}$ cells. Mass spectra were analysed according to the different masses of citrate. Data on 1-$^{13}C$-glutamine within the main text 1-13C-Gln is used ? labeling are summarized in Supplementary Fig. 5. Fractions of **d** m + 1 $^{15}N$ isotopologue in selected amino acids and **e** $^{15}N$-enriched nucleobases in unstimulated and IL10 + CpG-stimulated P493-6 MYC$^{low}$, as well as P493-6 MYC$^{high}$ cells. **f,g** Relative cell counts of IL10 + CpG-stimulated P493-6 MYC$^{low}$ and P493-6 MYC$^{high}$ cells grown in Gln-deprived and metabolite-treated media. Cells were treated with either **f** TCA metabolites (Pyr = pyruvate, α-KG = dimethyl 2-oxoglutarate, Mal = diethyl malate, OAA = oxaloacetate) or **g** aspartate (Asp) and nucleobases (A = adenine, T = thymine) for 24 h. For all plots, error bars represent mean ± SD of three independent experiments and results from Bonferroni post hoc tests on a one-way ANOVA results are given (*$p < 0.05$, **$p < 0.01$, ***$p < 0.001$)

distinction between DLBCL subtypes and a model for HL cells was provided[15,16]. However, the precise network of factors regulating the different metabolic subtypes remains unclear. It is also uncertain how the microenvironment in lymphomas may be involved in metabolic reprogramming and whether glutaminolysis in this context is similar to MYC-dependent lymphoma[17,18]. Inhibition of glutaminolysis is considered a promising therapeutic option in various cancers[19]. It is therefore vital to understand its role and mechanism of deregulation in lymphomas.

Here we hypothesize, that Gln metabolism is important for lymphoma proliferation and regulated by a combination of intracellular signaling pathways such as STAT3 and NF-κB, either activated by factors of the microenvironment or intrinsic genetic lesions. To investigate this, an integrative analysis of transcriptomic, metabolomic, and clinical data was conducted, to identify and delineate metabolic processes and the regulation of genes involved in glutaminolysis that may support B-cell proliferation and transformation specifically.

## Results

**Metabolic adaptation by IL10 and CpG in MYC-deprived B-cells**. Various studies have revealed that the adaptation of metabolic processes is essential to sustained proliferation. We recently analyzed the capacity of different B-cell-stimulating factors to induce proliferation in MYC-deprived cells[12]. To evaluate the specific changes in metabolism and global GE induced by these factors, P493-6 cells were treated with either CD40L, CpG, αIgM (BCR crosslink), IGF1, interleukin (IL)10, or various dual combinations thereof for 24 h. Linear regression analysis revealed that the activation of MYC-deprived P493-6 (MYC$^{low}$) cells by a combination of IL10 and CpG was associated with synergistic changes in cell metabolism and global activation of GE (Fig. 1 and

Supplementary Fig. 1). Given this synergy and our recent observation that IL10R and Toll-like receptor (TLR) 9 drive cell proliferation through the activation of STAT3 and NF-κB signaling, we focused our investigation on the activation of these signaling pathways[13]. The corresponding changes in intracellular metabolites and related genes in IL10-, CpG-, and IL10 + CpG-stimulated MYC$^{low}$ cells are presented in Fig. 1. The intracellular amounts of TCA cycle intermediates and amino acids are strongly increased by IL10 + CpG co-stimulation compared with MYC$^{low}$ cells exposed to either or no stimulus (Fig. 1a). These metabolic changes were accompanied by a marked increase in the expression of genes involved in glutaminolysis and glycolysis (Fig. 1b,c). Importantly, total cellular protein amounts were also increased (Supplementary Data 1). Interestingly, IL10 + CpG co-stimulation of MYC$^{low}$ cells resulted in increased cell size similar to that of MYC$^{high}$ cells (Supplementary Fig. 2).

Next, we measured both uptake of Gln and glucose (Glc), and secretion of lactate to determine the lactate-to-glucose molar ratio (Lac/Glc) (Supplementary Fig. 3). Strikingly, the uptake of both Gln and Glc was significantly increased in IL10 + CpG-stimulated MYC$^{low}$ cells compared with cells exposed to either or no stimulus and even more so in MYC$^{high}$ cells (Supplementary Fig. 3a, b). The Lac/Glc ratio was similar across the conditions analyzed and indicated that Glc is predominantly converted to lactate as reported before (Supplementary Figure 3c)[20]. However, compared with MYC$^{high}$ cells, both Glc consumption and lactate production are lower in IL10 + CpG-stimulated MYC$^{low}$ cells.

To test whether increased glycolysis and/or glutaminolysis induced by IL10R and TLR9 signaling was necessary to support B-cell proliferation, we cultured P493-6 cells in the presence or absence of Gln or Glc. Cell doubling was monitored over 3 days.

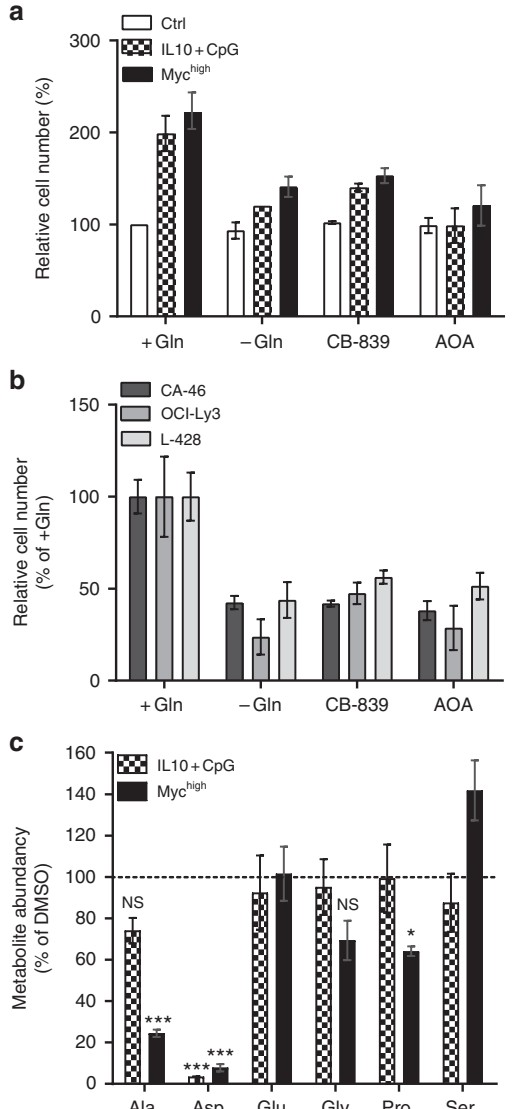

**Fig. 3** Proliferation of P493-6 and B-cell lymphoma cells requires transaminase activity. **a** Inhibition of cell doubling of IL10 + CpG-stimulated P493-6 MYC$^{low}$ and P493-6 MYC$^{high}$ cells after 24 h of treatment with CB-839 or AOA. **b** Inhibition of cell doubling of the B-cell lymphoma cell lines CA-46, OCI-Ly3, and L-428 using CB-839 or AOA treatment for 24 h. **c** Intracellular amino acid abundance in IL10 + CpG-stimulated MYC$^{low}$ and MYC$^{high}$ cells 24 h after AOA treatment and stimulation. For each amino acid data were normalized relative to the corresponding values of DMSO-treated cells. For all plots, error bars represent mean ± SD of three independent experiments and results from Bonferroni post hoc tests on a one-way ANOVA are given ($^*p < 0.05$, $^{**}p < 0.01$, $^{***}p < 0.001$)

In media containing both Glc and Gln, the proliferation of IL10 + CpG-stimulated MYC$^{low}$ cells was comparable to that of MYC$^{high}$ cells (Fig. 1d,e). Withdrawal of Gln strongly reduced cell doubling in both IL10 + CpG-stimulated MYC$^{low}$ and MYC$^{high}$ cells, whereas the effect of Glc removal was less pronounced (Fig. 1d,e). Notably, neither unstimulated nor singly stimulated MYC$^{low}$ cells proliferated under the experimental conditions used (Supplementary Fig. 4).

In conclusion, stimulation of P493-6 MYC$^{low}$ cells with IL10 + CpG resulted in global metabolic reprogramming, with Gln being essential for sustaining proliferation.

**Anabolic processes of glutaminolysis in B-cell proliferation.** To further explore the role of Gln in the proliferation of IL10 + CpG-stimulated MYC$^{low}$ cells, we performed $^{13}$C-Gln tracing experiments. Cells were cultured in the presence of $^{13}$C$_5$-Gln for 24 h, the fractional abundance of $^{13}$C-isotopologues of selected metabolites was measured and the isotopic enrichment calculated (Fig. 2a). In comparison with unstimulated MYC$^{low}$ cells, IL10 + CpG-stimulated MYC$^{low}$ cells were significantly enriched with $^{13}$C-labelled TCA cycle intermediates, thus corroborating the importance of Gln as an anabolic precursor. A comparable inclusion of labelled carbons into the TCA cycle was observed in MYC$^{high}$ cells, consistent with previously published data on the P493-6 cell line[8]. More importantly, we observed increased incorporation of Gln-derived carbons into the amino acids aspartic acid (Asp), glutamic acid (Glu), proline (Pro), and the non-proteinogenic amino acid ornithine, but not into other amino acids analysed (e.g., alanine, glycine, and serine) (Fig. 2a). Hence, IL10 + CpG-stimulated B-cells utilize Gln-derived carbons comparable to B-cells aberrantly expressing MYC. However, the fractional abundance of m + 4 citrate indicates that IL10 + CpG-stimulated cells use less Gln for oxidative phosphorylation than MYC$^{high}$ cells, whereas reductive carboxylation of Gln-derived α-KG to citrate (m + 5 citrate) seems to be equally present in both IL10 + CpG-stimulated MYC$^{low}$ and MYC$^{high}$ cells (Fig. 2b,c). To further support our conclusion on reductive carboxylation, cells were labeled with 1-$^{13}$C-Gln for 24 h. Measurement of the m + 1 citrate $^{13}$C-isotopologue fraction revealed that IL10 + CpG-stimulated MYC$^{low}$ cells contained a higher fraction of m + 1 citrate than unstimulated cells (9.7% vs. 5.4%) (Supplementary Fig. 5a, b). In comparison, the respective fraction of m + 1 citrate in MYC$^{high}$ cells was 12% (Supplementary Fig. 5a, b).

To gain further insight into Gln catabolism in IL10 + CpG-stimulated MYC$^{low}$ cells, α-$^{15}$N-Gln tracing studies were performed (Fig. 2d). The amino acids Ala, Asp, Glu, and Pro showed increased enrichment of $^{15}$N in IL10 + CpG-stimulated MYC$^{low}$ cells. Even stronger was the incorporation in MYC$^{high}$ cells. Importantly, labeled nitrogen derived from Asp and Gly was also observed in nucleobases (m + 1, m + 2) (Fig. 2e). The very low levels of nitrogen incorporation in the letter can be explained by the used tracer, where only the nitrogen in α-position of Gln was labeled to assess the contribution to nucleobase biosynthesis.

In summary, B-cell proliferation induced by the co-stimulation with IL10 and CpG relies on glutaminolysis and Gln-dependent anabolic processes.

**Glutaminolysis supports anabolic functions after IL10+CpG.** To further evaluate which of the Gln-derived metabolites are essential for cell proliferation, we added various metabolites to Gln-starved IL10 + CpG-stimulated MYC$^{low}$ and unstimulated MYC$^{high}$ cells, respectively, and evaluated their capacity to restore cell doubling (Fig. 2f). Of the TCA-derived metabolites tested (malate, oxaloacetate (OAA), α-KG, and pyruvate (Pyr)), only OAA restored the proliferation of IL10 + CpG-stimulated cells significantly. In contrast, the proliferation of MYC$^{high}$ cells was restored in part by Pyr and, to a much greater extent, by the addition of dimethyl α-KG, a cell-permeating form of α-KG (Fig. 2f).

The TCA cycle intermediate OAA provides the carbon skeleton for the production of aspartate in a glutamate-dependent reaction catalyzed by the aminotransferase GOT2. Hence, next we tested whether supraphysiological levels of Asp were also able to restore cell proliferation following Gln deprivation. Indeed, Asp restored the proliferation of Gln-starved IL10 + CpG-stimulated cells, but surprisingly not of Gln-deprived MYC$^{high}$ cells (Fig. 2g). As Asp

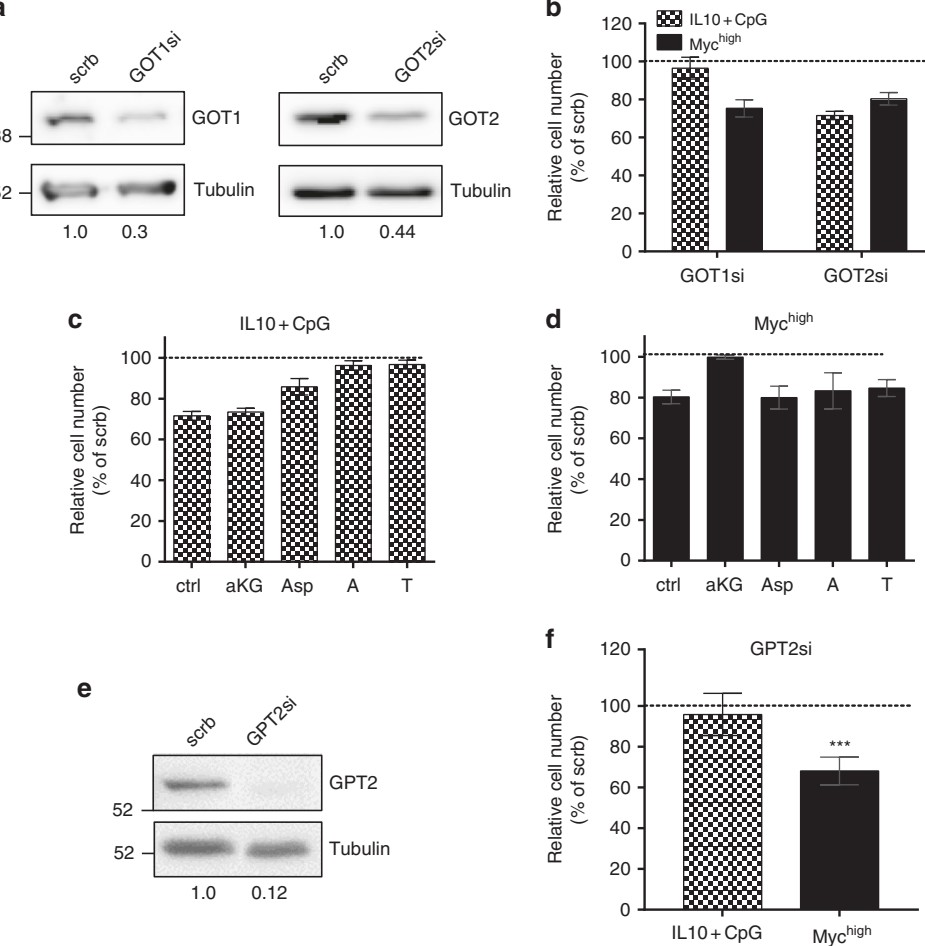

**Fig. 4** GOT2 supports the proliferation of IL10 + CpG-stimulated and MYC-overexpressing cells. **a** Transient knockdown of *GOT1* and *GOT2* in P493-6 cells was performed by siRNA transfection. Knockdown efficiencies (GOT*/tubulin) relative to scrb control are provided under the images. **b** Relative cell numbers of IL10 + CpG-stimulated P493-6 MYC$^{low}$ and P493-6 MYC$^{high}$ after *GOT1* or *GOT2* knockdown depicted in **a**. Relative cell numbers of **c** IL10 + CpG-stimulated P493-6 MYC$^{low}$ and **d** unstimulated P493-6 MYC$^{high}$ cells after *GOT2* knockdown and indicated α-KG, Asp, A or T addition as described in Fig. 2. **e** Transient knockdown of *GPT2* in P493-6 cells performed by siRNA transfection. Knockdown efficiencies (GPT2/tubulin) relative to scrb control are provided under the images. **f** Relative cell numbers of IL10 + CpG-stimulated P493-6 MYC$^{low}$ and P493-6 MYC$^{high}$ cells after *GPT2* knockdown as depicted in **e**. All experiments were performed in triplicates. For **b–d** and **f**, error bars represent mean ± SD of three independent experiments and results from Bonferroni post hoc tests on a one-way ANOVA are given (*$p < 0.05$, **$p < 0.01$, ***$p < 0.001$)

is an important precursor in nucleotide biosynthesis, we further investigated the capacity of nucleobases to restore cell proliferation. Addition of the nucleobases adenine (A) or thymine (T) led to the almost complete restoration of IL10 + CpG-induced MYC$^{low}$ cell proliferation (Fig. 2g). As demonstrated for Asp, the nucleobases adenine or thymine also proved unable to overcome the Gln deficiency in MYC$^{high}$ cells in culture medium. This indicates that Gln is essential for nucleotide biosynthesis in IL10 + CpG-stimulated cell proliferation and suggests that IL10 and CpG together reprogram glutaminolysis of resting B-cells toward anabolic metabolism to support Asp and nucleobase synthesis, and thus cell proliferation.

As OAA, Pyr, and α-KG are capable of replenishing the TCA cycle, which provides succinate and reducing equivalents to oxidative phosphorylation, we compared the oxygen consumption (OCR) and extracellular acidification rates (ECAR) in the presence or absence of Gln (Supplementary Figure 6a, b). In IL10 + CpG-stimulated MYC$^{low}$ cells the OCR and ECAR were higher than in unstimulated MYC$^{low}$ cells but remained unchanged in the absence of Gln. In contrast, the OCR and ECAR rates of MYC$^{high}$ cells were reduced significantly to the level of unstimulated MYC$^{low}$ cells upon Gln withdrawal. It is therefore

reasonable to argue that IL10 + CpG-stimulated MYC$^{low}$ cells use substrates other than Gln to support respiration, whereas Gln is necessary to maintain OCR in MYC$^{high}$ cells. This is consistent with the above observation that IL10 + CpG-stimulated MYC$^{low}$ cells use less Gln for oxidative phosphorylation than MYC$^{high}$ cells (Fig. 2). In addition, OCR was measured after the respective inhibition of the mitochondrial pyruvate carrier (MPC1) by UK5099, carnitine palmitoyltransferase (CPT1), a key regulator of the β-oxidation of long-chain fatty acids, by etomoxir, and glutaminolysis by the GLS inhibitor Bis-2-(5-phenylacetamido-1,3,4-thiadiazol-2-yl)ethyl sulfide (BPTES) (Supplementary Fig. 6c). As expected, BPTES affected MYC$^{high}$ cells much more than IL10 + CpG-stimulated MYC$^{low}$ cells. MYC$^{low}$ and IL10 + CpG-stimulated MYC$^{low}$ cells consumed less amounts of fatty acids than MYC$^{high}$ cells. Furthermore, in contrast to both unstimulated MYC$^{low}$ and MYC$^{high}$ cells, oxidative phosphorylation in IL10 + CpG-stimulated cells was almost independent of Glc.

Thus, our data suggest that IL10 + CpG-stimulated B-cells are characterized by glutaminolysis supporting anabolic functions, and oxidative phosphorylation is supported by both fatty acids and Gln.

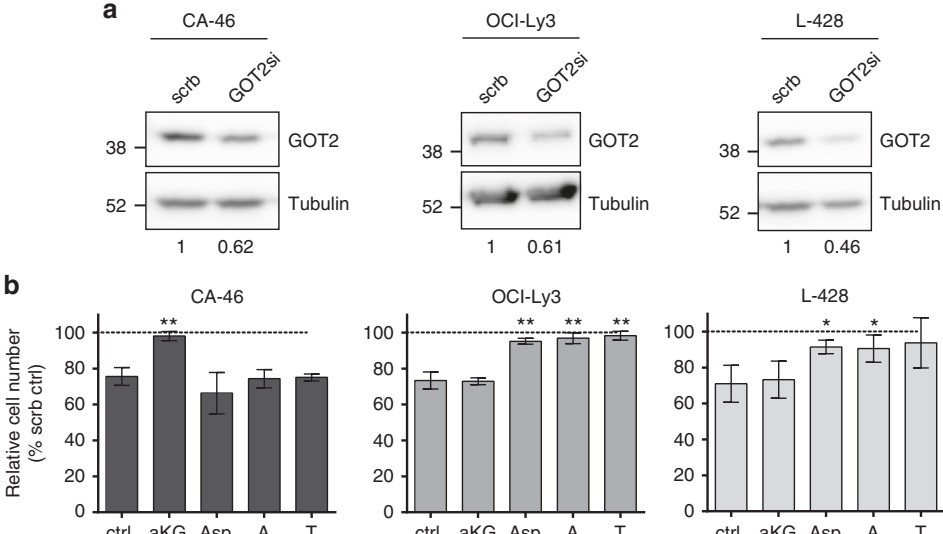

**Fig. 5** Context dependent role of GOT2 in lymphoma cells. **a** Immunoblot of GOT2 in CA-46, OCI-Ly3, and L-428 lymphoma cells 24 h after siRNA-mediated knockdown. Knockdown efficiencies (GOT2/tubulin) relative to scrb control are presented under the images. **b** Relative cell numbers of CA-46, OCI-LY3, and L-428 cells after *GOT2* knockdown and metabolite treatment. Metabolites were used as described in Fig. 2. Error bars represent mean ± SD of three independent experiments and results from Bonferroni post hoc tests on a one-way ANOVA are given (*$p < 0.05$, **$p < 0.01$, ***$p < 0.001$)

**Transaminases in IL10 + CpG-mediated metabolic reprogramming**. Glutaminolysis and cancer cell proliferation have been targeted in various studies[6,8,21]. The importance of the aspartate aminotransferases GOT1 and GOT2 in the generation of Glu and Asp from α-KG and OAA, respectively, prompted us to study whether their inhibition may suppress cell proliferation as readily as the inhibition of GLS. As shown above, both Asp and α-KG restored cell proliferation in a context dependent manner. This implies different, context-dependent roles of transaminases in glutaminolysis and B-cell proliferation. We therefore compared the effects of Gln deprivation, GLS inhibition, and transaminase inhibition on proliferation of IL10 + CpG-stimulated MYC$^{low}$ and unstimulated MYC$^{high}$ cells. Inhibitors such as CB-839 for GLS or aminooxyacetate (AOA) for transaminases were used for this purpose. AOA is a general inhibitor of pyridoxal phosphate-dependent enzymes, including transaminases, which are involved in amino acid metabolism. Furthermore, it displayed significant antitumoral effects in preclinical studies[9]. Treatment of cells with CB-839 or AOA for 24 h caused a strong reduction in cell proliferation in both IL10 + CpG-stimulated MYC$^{low}$ and unstimulated MYC$^{high}$ cells (Fig. 3a). The observed decrease was comparable to that caused by Gln starvation. Similar effects were also observed for lymphoma cell lines representative of aberrant MYC activity (CA-46) or constitutive Jak/STAT and NF-kB pathway activation (OCI-Ly3, L-428) (Fig. 3b). The similar effects of Gln deprivation and AOA treatment were also demonstrated for the additional B-cell lines BJAB, TMD-8, HBL-1, and KM-H2, thus underscoring the importance of transaminases in sustaining B-cell lymphoma proliferation (Supplementary Fig. 7b).

Importantly, measurements of intracellular amino acid amounts revealed that Asp levels dropped significantly after AOA treatment in both IL10 + CpG-stimulated MYC$^{low}$ and unstimulated MYC$^{high}$ cells (Fig. 3c). The intracellular levels of other amino acids such as Glu, Gly, Pro, or Ser were only slightly affected by transaminase inhibition. Notably, Ala levels were only significantly affected after AOA treatment in MYC$^{high}$ cells (Fig. 3c).

To explore whether the loss of Asp and nucleotide biosynthesis can explain the proliferation block by AOA treatment, we performed again proliferation restoration experiments

(Supplementary Fig. 8a, b). Consistent with the results for Gln deprivation, the proliferation of IL10 + CpG-stimulated MYC$^{low}$ cells was restored significantly by Asp or nucleobase addition following AOA treatment. The same held true for OCI-Ly3 and L-428 cells. Furthermore, only a marginal but insignificant increase in MYC$^{high}$ cell proliferation was observed after Asp or nucleobase supplementation in line with the effects of Gln deprivation and this was also observed in CA-46 cells. Using $^{13}C_5$-Gln as a tracer, glutaminolysis was further characterized in the lymphoma cell lines OCI-Ly3, L-428, and CA-46. Cells were labeled with $^{13}C_5$-Gln for 24 h and the fractional abundance of $^{13}C$ isotopologs of citrate was determined (Supplementary Fig. 5c). As for the P493-6 cells, the m + 4 isotopolog was the most abundant in CA-46 and OCI-Ly3 cells. In L-428 cells, in contrast, the m + 5 isotopolog dominated.

To further elucidate the impact of this transaminase inhibition on energy production, we measured the OCR and ECAR following AOA treatment in P493-6 cells. The cells were seeded in Gln-depleted media, and Gln and AOA were added consecutively. OCR was monitored continuously (Supplementary Fig. 8c, d). In the absence of Gln, IL10 + CpG-stimulated cells displayed greater OCR than MYC$^{high}$ cells. Addition of Gln increased the OCR of MYC$^{high}$ cells but not of IL10 + CpG-stimulated MYC$^{low}$ cells. Importantly, this effect was reversed by AOA treatment. However, the OCR of IL10 + CpG-stimulated MYC$^{low}$ cells remained unaffected by AOA treatment.

In conclusion, transamination is critical in sustaining proliferation in both IL10 + CpG-stimulated MYC$^{low}$ and unstimulated MYC$^{high}$ cells. However, the former, unlike MYC$^{high}$ cells, are not dependent on transamination for oxidative phosphorylation.

**GOT2 in IL10 + CpG-stimulated MYC$^{low}$ and MYC$^{high}$ cells**. The transaminase GOT2 catalyzes the conversion of Glu and OAA to α-KG and Asp, whereas GOT1 catalyzes the reverse reaction in the malate–aspartate shuttle as long as the mitochondrial electron transport chain is functional[19,21–23].

The abundance and catalytic activity of GOTs is increased after IL10 + CpG stimulation of MYC$^{low}$ cells as shown in Supplementary Data 1 and Supplementary Fig. 9. Thus, the

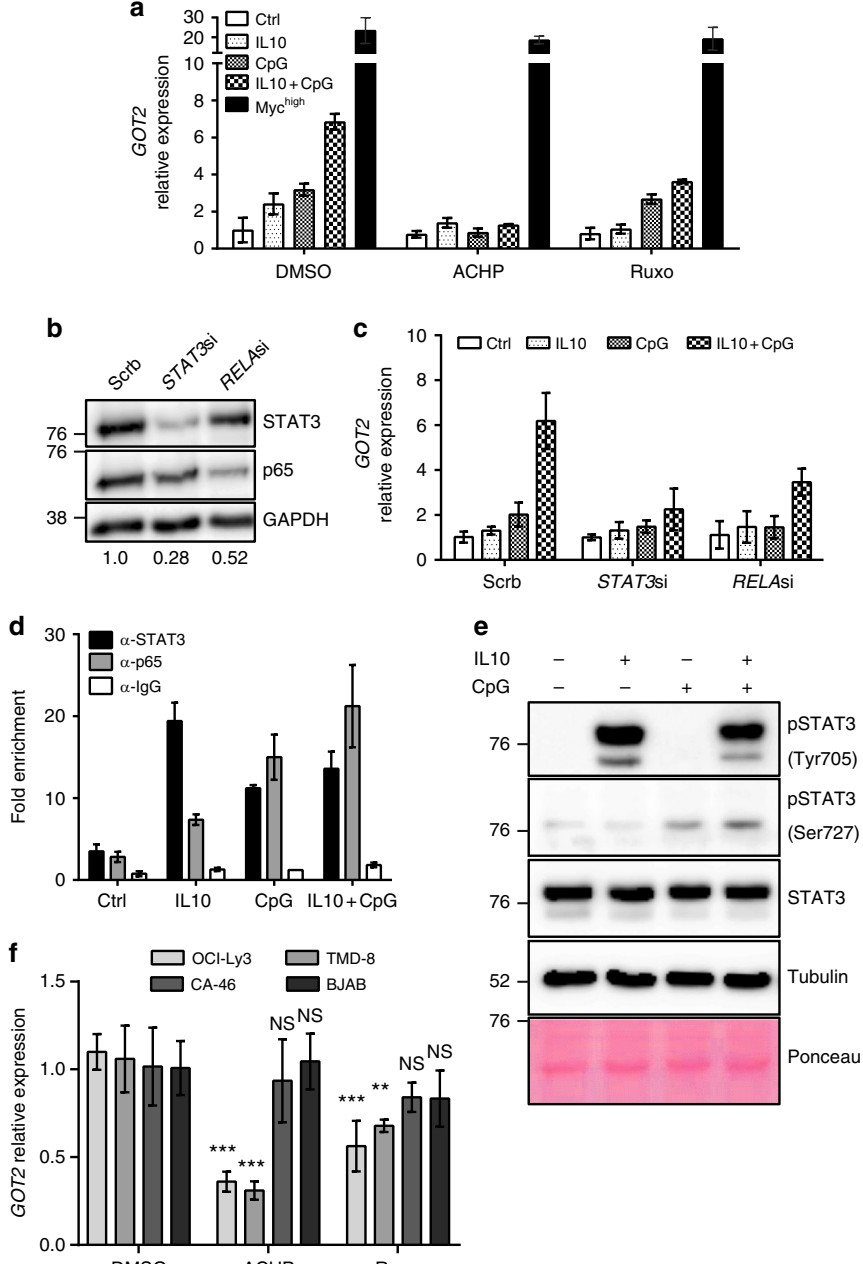

**Fig. 6** *GOT2* expression in B-cells is up-regulated by NF-κB/STAT3 activation or MYC. **a** *GOT2* mRNA expression in 24h-stimulated P493-6 MYC[low] and unstimulated P493-6 MYC[high] cells after inhibition of NF-κB signalling by ACHP or JAK/STAT signalling by Ruxolitinib. **b** Representative immunoblot of STAT3 and p65 protein levels 24 h after *STAT3* (*STAT3si*) or *RELA* siRNA (*RELAsi*) transfection of P493-6 cells. Knockdown efficiencies (protein/GAPDH) relative to scrb control are presented under the images. **c** *GOT2* mRNA expression in 24h-stimulated P493-6 MYC[low] cells after siRNA transfection shown in **b**. **d** Chromatin immunoprecipitation of STAT3 and p65 1 h after stimulation of P493-6 MYC[low] cells. Displayed are average qPCR results and technical errors (SD) of the *GOT2* promoter (−20 bp) from a representative stimulation out of three. Enrichment was calculated relative to an inactive control region (*PRAME*). **e** Full tyrosine and serine phosphorylation of STAT3 is only achieved after IL10 + CpG costimulation as revealed by immunoblot analysis. Cells were stimulated with IL10, CpG or both for 30 min. **f** *GOT2* mRNA expression of TMD-8 and OCI-Ly3 cells in comparison with BJAB and CA-46 cells 24 h after ACHP or rRuxolitinib treatment. For additional details about the used cell lines also refer to Supplementary Fig. 7a. With the exception of **d**, all values are given as mean ± SD of three independent replicates. Results from Bonferroni post hoc test on a two-way ANOVA are presented (***$p < 0.001$, **$p < 0.01$)

transcriptional activation (Fig. 1c) is also translated into protein and enzymatic activity.

To investigate whether GOT1 and GOT2 are involved in the Gln-dependent proliferation of B-cells, we performed RNA interference (RNAi)-mediated knockdowns of both the cytosolic GOT1 and the mitochondrial GOT2 in P493-6 cells (Fig. 4a). A knockdown of about 60%–70% was achieved for both. The achieved knockdown of *GOT2* was sufficient to cause a reduction

in cell proliferation of both IL10 + CpG-stimulated MYC[low] and unstimulated MYC[high] cells. The *GOT1* knockdown, on the other hand, had no discernible effect on the proliferation of IL10 + CpG-stimulated MYC[low] cells, whereas proliferation in MYC[high] cells decreased to the same extent as seen with the *GOT2* knockdown (Fig. 4b).

To investigate the capacity of Asp or α-KG to restore reduced cell proliferation after *GOT2* knockdown, cells were grown in

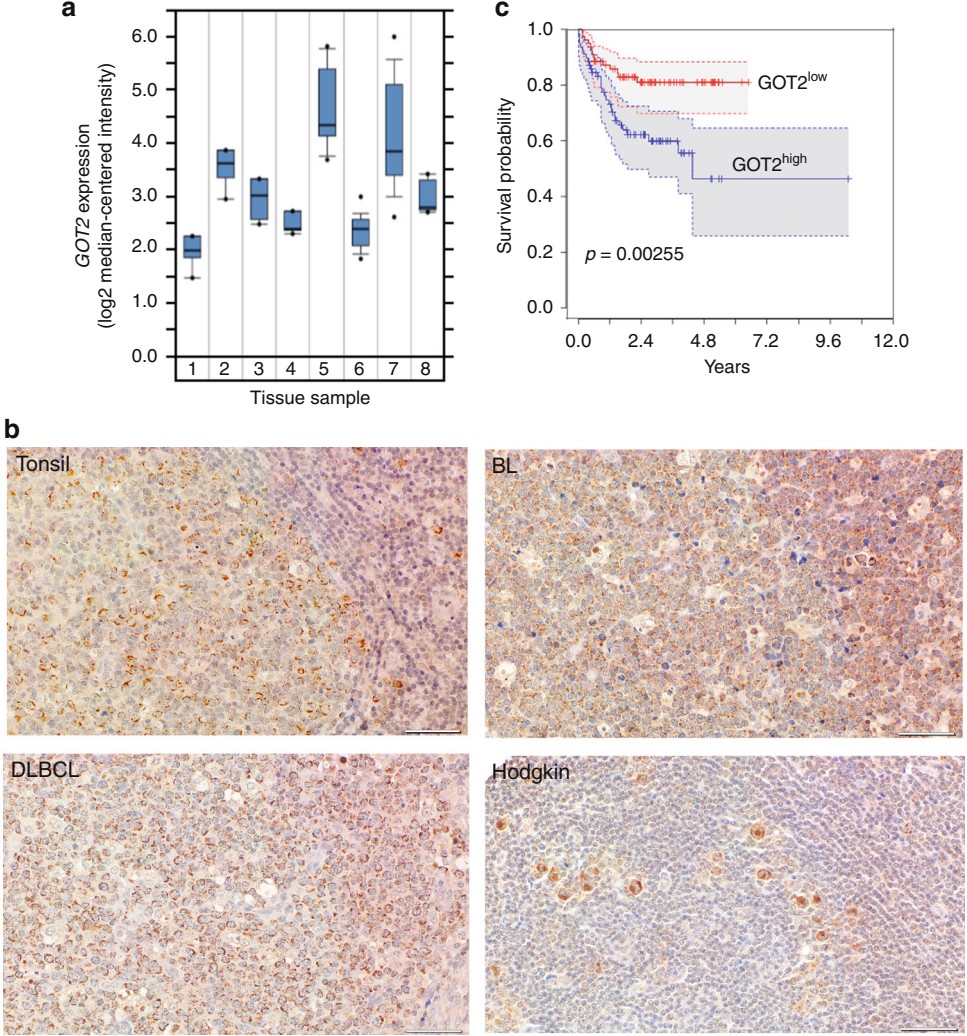

**Fig. 7** *GOT2* is overexpressed in lymphoma subtypes and correlates with worse overall survival in DLBCL patients. **a** Microarray gene expression data of *GOT2* obtained at *ONCOMINE*[68,69]. Expression of normal B-cell controls (1 = B lymphocytes ($n = 5$), 2 = centroblasts ($n = 5$), 3 = memory B -cells ($n = 5$), 4 = naive pre-germinal B- cells ($n = 5$)), and lymphoma samples (5 = BL ($n = 17$), 6 = CLL ($n = 34$), 7 = DLBCL ($n = 32$), and 8 = MCL ($n = 8$)) are given. Boxes are showing median (middle line), upper and lower quantile (edges), minimum and maximum (bars), and outliners (dots). **b** Immunohistological staining of GOT2 in representative samples of tonsils, BLs, DLBCLs, and HLs (see also Table 1). The scale bar represents 50 μm. In Supplementary Fig. 12, a corresponding GOT2 staining for the liver, lung, muscles, and skin is provided showing a homogeneous *GOT2* expression in hepatocytes. **c** Kaplan–Meier curves of DLBCL patients ($n = 157$) treated with R-CHOP[18]. Patients were grouped according to their *GOT2* expression relative to the median of the group (high > median and low ≤ median). Additional details using a Cox proportional hazards regression model to estimate the clinical outcome of patients with DLBCL in relation to the expression of *GOT2* are presented within Supplementary Table 1

normal medium and cell proliferation was analyzed after addition of α-KG, Asp, or nucleobases. Importantly, the addition of adenine and thymine fully restored the proliferation of IL10 + CpG-stimulated MYC$^{low}$ cells, while Asp restored it partially (Fig. 4c). In contrast, MYC$^{high}$ cell proliferation could only be restored upon addition of α-KG (Fig. 4d). Differences between the effects of AOA treatment and *GOT2* knockdown indicated the involvement of other transaminases. This prompted us to investigate the glutamate pyruvate transaminase (GPT2), the GE of which had increased less in IL10 + CpG-stimulated MYC$^{low}$ cells compared with that in MYC$^{high}$ cells. No significant decrease in cell proliferation was observed after *GPT2* knockdown in IL10 + CpG stimulated MYC$^{low}$ cells. In contrast, cell proliferation of MYC$^{high}$ cells was decreased after *GPT2*

knockdown (Fig. 4e/f). A specific role for GPT2 in MYC$^{high}$ cells is in line with the above described higher nitrogen incorporation into Ala and the decrease in Ala abundance after AOA treatment and the recent description of cells with cooperating oncogenic mutations[24].

In order to gain insight into the role of GOT2 in B-cell lymphoma, OCI-Ly3, L-428, and CA-46 cells were selected. Reasonable *GOT2* knockdown was achieved in all three cell lines (Fig. 5a). Knockdown of *GOT2* significantly decreased the proliferation of all cell lines (Fig. 5a,b). However, as in IL10 + CpG-stimulated P493-6 MYC$^{low}$ cells, the reduced cell proliferation was almost completely restored by Asp or nucleobases. In contrast, the reduced proliferation of CA-46 cells following *GOT2* knockdown was restored only by the addition of α-KG but not by

**Table 1 Immunohistochemical analysis of GOT2 distribution in B-cell lymphoma**

|  | Number of cases | Positive | Partial positive |
|---|---|---|---|
| Hodgkin lymphoma | 15 | 15 | 0 |
| Diffuse large B-cell lympoma | 10 | 8 | 2 |
| Burkitt lymphoma | 6 | 6 | 0 |
| Mediastinal B-cell lymphoma | 4 | 4 | 0 |
| Acute B-cell leukemia | 5 | 2 | 3 |

Partially positive ($\leq 50\%$ of lymphoma cells are stained with anti-GOT2 antibody) and positive ($> 50\%$ of lymphoma cells are positive for GOT2). For additional details, see also Fig. 7B and Suppl. Material and Methods section.

Asp or nucleobases. It thus appears that glutaminolysis and cell proliferation in CA-46 cells depend on GOT2, as observed in P493-6 MYC[high] cells.

In summary, we propose a model in which GOT2 is a metabolic hub in B-cells in a context-dependent manner, providing the cells with either Asp for biosynthesis or α-KG for energy demands.

**GOT2 expression is regulated by a combination of STAT3/NF-κB.** Previously, we demonstrated that IL10 and CpG activate STAT3 and NF-κB signaling in P493-6 cells and cause a synergistic increase in the expression of the cell cycle regulator *CDK4* through direct, simultaneous binding to its gene promoter[12]. To address the question, whether IL10 + CpG stimulation also increases the expression of *GOT2* in a NF-κB and/or STAT3-dependent manner, we treated MYC[low] cells with the IKK inhibitor ACHP and the JAK inhibitor Ruxolitinib before stimulation with IL10, CpG, or IL10 + CpG (Fig. 6a). Ruxolitinib treatment decreased *GOT2* expression in IL10 + CpG-stimulated cells comparable to that observed in cells stimulated with CpG only. In contrast, ACHP treatment completely abrogated the increase in *GOT2* expression after any stimulation. Notably, the increase in *GOT2* expression in MYC[high] cells was not affected by any of the inhibitors (Fig. 6a).

For further validation, we performed RNAi-mediated *STAT3* and *RELA* (p65) knockdowns in MYC[low] cells before IL10 and CpG stimulation (Fig. 6b). Consistent with the inhibitor treatments, both *STAT3* and *RELA* knockdowns resulted in a strong reduction in *GOT2* expression in IL10 + CpG-stimulated MYC[low] cells (Fig. 6c). Furthermore, a direct regulatory function of STAT3 and p65 in *GOT2* expression was observed by corresponding chromatin immunoprecipitation (ChIP) (Fig. 6d). In MYC[low] cells stimulated with IL10 or CpG, we observed an increase in the binding of STAT3 or p65/NF-κB to the proximal promoter of *GOT2*. However, transcription factor binding exerted little effect on gene transcription. Importantly, following IL10 + CpG-co-stimulation, strong joint binding of STAT3 and p65 to the same promoter region was observed. This resulted in marked *GOT2* expression. This is the first ever demonstration that *GOT2* is a direct target of STAT3 and p65 NF-κB.

In addition, analysis of STAT3 phosphorylation in MYC[low] cells showed that IL10 and CpG mediated the respective phosphorylation of Tyr-705 and Ser-727 (Fig. 6e). Hence, both coordinated STAT3 phosphorylation, which is a known prerequisite for the interaction of STAT3 with p65, and the joint binding of STAT3 and p65 NF-κB to their target genes mechanistically support the synergy of IL10 and CpG (Fig. 6e)[25].

To exclude that increased cell proliferation of IL10 + CpG stimulated MYC[low] cells was rather the result of increased *MYC* expression mediated by STAT3 and p65 activation, both changes in Myc protein abundance and the effect of RNAi-mediated Myc

knockdown onto cell proliferation were monitored over 24 h (Supplementary Fig. 10). A transient low activation of Myc was observed after both CpG and IL10 + CpG stimulation of MYC[low] cells. However, cell proliferation was only increased upon IL10 + CpG costimulation (Supplementary Fig. 10a). Furthermore, the knockdown of Myc affected the proliferation of MYC[high] cells much stronger, despite a less pronounced reduction in Myc (Supplementary Figure 10b, c). Considering STAT3/p65 ChIP analysis, Jak/STAT and NF-κB inhibition, and Myc knockdown data, we conclude that the synergistic effect of IL10 and CpG is driven mainly by STAT3 and NF-κB and not by Myc.

To test whether *GOT2* expression depends on STAT3 and NF-κB in other cell lines as well, OCI-Ly3 and TMD-8 lymphoma cells were compared with BJAB and CA-46 cells, which do not depend on STAT3/NF-κB (Supplementary Figure 7a). OCI-Ly3 and TMD-8 cells were derived from DLBCL and display constitutively active STAT3 and NF-κB signaling due to constitutively active TLR signaling and autocrine cytokine production[26]. A significant reduction in *GOT2* expression was observed on treating OCI-Ly3 and TMD-8 cell lines with Ruxolitinib and ACHP (Fig. 6f).

It is thus reasonable to suggest that a general mechanism of *GOT2* expression has been identified which relies on combined Jak/STAT and NF-κB signaling activity. Furthermore, our data support a model in which the simultaneous activity of STAT3 and NF-κB may substitute for the transcriptional activity of aberrant *MYC* expression in regulating *GOT2* expression and, consequently, B-cell glutaminolysis and proliferation.

**Aberrant *GOT2* expression in lymphoma subtypes is prognostic.** Next we compared *GOT2* GE between normal B-cell controls (naive pre-germinal B-cells, B lymphocytes, centroblasts, and memory B-cells) and different B-cell lymphomas (BL, DLBCL, chronic lymphocytic leukemia (CLL) and mantle cell lymphoma (MCL)). Among controls, *GOT2* GE was highest in proliferating centroblasts. Even higher *GOT2* expression was found in DLBCL ($p = 7.60E-7$, $n = 32$) and BL ($p = 4.94E-9$, $n = 17$) samples, but not in CLL and MCL in relation to normal B-cell subsets (Fig. 7a). Nearly all BL cases (14/17) expressed greater levels of *GOT2* than centroblasts, whereas 14/32 DLBCL cases demonstrated increased expression of *GOT2* (Supplementary Figure 11a, b). Importantly, no significant differences in GE were observed for *GLS1*, *GLS2*, *GLUD1*, *GLUD2*, *GOT1*, and *GPT1* in this dataset (Supplementary Figure 11c-h). We therefore conclude that *GOT2* is expressed aberrantly in a subset of lymphoma and is characteristic of BL and a subgroup of DLBCL.

Next, GOT2 was analysed using immunohistochemistry (Table 1, Fig. 7b, and Supplementary Fig. 12). We observed that liver samples were positively stained for GOT2, whereas other tested tissues (muscle, skin, and lung) were negative. Inside the lymph nodes, normal lymphocytes are usually not stained with

anti-GOT2 antibodies, except for a few cells in the germinal center (Fig. 7b, tonsil). Lymphoma tissues of HL, DLBCL, and BL patients, in contrast, stained strongly for GOT2 thus supporting the GE data (Fig. 7a, b).

To evaluate whether aberrant *GOT2* GE affects overall survival, we analysed a cohort of 157 DLBCL patient that had been treated with immunochemotherapy (R-CHOP (Rituximab, Cyclophosphamide, Hydroxydaunomycin, Oncovin, Prednisone))[17]. The Kaplan–Meier plot demonstrates that high aberrant *GOT2* expression is significantly associated with a shorter overall survival ($p = 0.003$) (Fig. 7c). The inclusion of *GOT2* expression in a Cox proportional hazards model, comprising in addition the age, the international prognostic index (IPI), and the molecular classification as activated B-cell (ABC)-like DLBCL revealed an enhanced relative risk in patients with high *GOT2* expression (hazard ratio: 2.28; $p = 0.03756$) (Supplementary Table 1)[13]. This indicates that high *GOT2* expression is an independent prognostic factor.

## Discussion

A combination of techniques has been used to study the effects of the B-cell-stimulating factors IL10 and CpG on metabolism and proliferation of *MYC*-deprived B-cells. Our analysis demonstrates an important role of glutaminolysis and, in particular, GOT2 in B-cells and subtypes of B-cell lymphoma. Importantly, the metabolic responses are mediated by coordinated STAT3 and p65/NF-κB activity. Furthermore, in STAT3/NF-κB-mediated responses, Gln is utilized for Asp and nucleotide synthesis to support B-cell proliferation. Thus, STAT3/p65-dependent upregulation of *GOT2* is a hitherto undescribed mechanism in the mediation of glutaminolysis during lymphocyte proliferation and transformation, albeit additional aminotransferases may be involved in the B-cell model cell line studied.

In line with our view on the cooperation of microenvironmental factors is the description of the costimulation of T-cells via the T-cell receptor and CD28 to support metabolic reprogramming and proliferation[27,28]. An important role for Glc metabolism during lymphocyte activation has already been demonstrated[29]. In previous studies, evidence was provided that T-cell (or B-cell) receptor activation, TLR signaling, or IL-7 stimulation increase Glc consumption in lymphocytes[1,3,30]. Interestingly, full glycolytic responses rely mostly on AKT-associated signaling pathway activities[1,30]. In contrast, glutaminolysis in lymphocytes is regulated by either Myc- or CD28-mediated activation of the ERK pathway[2,31]. The central role of glutaminolysis regulated by the combination of STAT/NF-κB, as described here, thus extends our knowledge of metabolic networks in lymphocytes.

STAT3 alone has been described as a regulator of glycolysis and the electron transport chain[32]. Furthermore, a role in the metabolic shift toward Gln-dependent tumor growth, invasion, and bioenergetics in ovarian cancer was observed[33]. In ErbB2-positive breast cancer cells, NF-κB was found to upregulate *GLS1*, thus acting as an oncogenic signal promoting Gln utilization[34]. Our study reveals that in B-cells the combined action of STAT3/p65 drives cell proliferation on a low MYC background comparable to aberrant *MYC* expression. Although to some extent the cooperation of STAT3 and NF-κB signaling has already been described in apoptosis, proliferation, and inflammation, our data emphasizes the critical role of the cooperation of both transcription factors in reprogramming glutaminolysis in B-cells.

STAT3 and p65 have already been shown to interact and cooperate in regulating GE[35,36]. However, no canonical STAT3 or NF-κB-binding sites within the proximal *GOT2* promoter have been described to date. We propose a model in which the dual binding of NF-κB and STAT3 at the proximal *GOT2* gene promoter includes physical interaction as described for *ICAM1*[36]. Importantly, our study shows that STAT3 and p65 together bind to sequences that differ from the canonical binding sites. In addition, IL10 and CpG modulate STAT3 activity via phosphorylation of the amino acid residues Tyr-705 and Ser-727 in STAT3. Therefore, it is reasonable to conclude that both coordinated STAT3 phosphorylation and joint binding of STAT3 and p65 NF-κB to their target genes are important in the observed pathway synergy of these microenvironmental factors. Future studies may reveal additional posttranslational modifications in both STAT3 and p65[25]. We propose that STAT3/p65-regulated *GOT2* expression is an alternative regulatory circuit to Myc[12,37]. However, other Myc-regulatory pathways for metabolic enzymes have also been identified, such as MYC/micro-RNA-regulated *GLS* expression or glutamine synthase promoter demethylation[7,38]. Future evaluation will be necessary to determine whether additional indirect mechanism can be observed for STAT3/NF-κB or whether both transcription factors are also involved in super-enhancer regulation as demonstrated for Myc or might have crosstalks with PI3K, 14-3-3[39–41].

The important role of Asp in B-cell metabolism and proliferation detected maintains some parallels to recent studies. There it was revealed that a crucial role of cellular respiration is to maintain redox homeostasis to support Asp and therefore nucleotide synthesis[22,42]. In addition, it was shown in glioblastoma that the conversion of Glu to Gln fuels de novo purine biosynthesis[43]. Another study demonstrated that mammary epithelial cells can switch from oxidative deamination of Glu to α-KG catalyzed by GLUD to the conversion of OAA into Asp by transamination of Gln catalyzed by GOTs[44]. In this scenario, cells are provided with energy and amino acids simultaneously. The different glutaminolysis networks identified herein in which IL10 + CpG-stimulated MYC[low] cells use less Gln for oxidative phosphorylation compared with MYC[high] cells fits well with previous data demonstrating that MYC-dependent osteosarcoma cells use Gln for cellular respiration[10]. Therefore, we propose that GOT2 serves as a metabolic hub, providing cells with Asp and/or α-KG for energy demands and biosynthetic processes such as nucleotide biosynthesis in a context-dependent manner. Further, transaminases other than GOT2 such as GPT2 may contribute to the observed differences. GPT2 promotes Gln-dependent anaplerosis by catalyzing the reversible addition of an amino group from glutamate to Pyr generating Ala and α-KG to fuel the TCA. It is believed that this provides a link between glycolysis, glutaminolysis, and TCA anaplerosis. Smith et al.[24] provided evidence for a model in which GPT2 is critical for the cancer phenotype including Gln-driven TCA anaplerosis and coupling of glycolytic Pyr output to Gln catabolism[24]. Interestingly, GPT2 was dispensable in not fully transformed cells similar as observed herein for IL10 + CpG stimulated MYC[low] cells. Thus it seems to be reasonable that the Warburg effect supports oncogenesis via GPT2-mediated coupling of Pyr production to Gln catabolism in MYC[high] cells but not IL10 + CpG stimulated MYC[low] cells[24]. It remains to be seen whether the observed oxidative decarboxylation of malate into Pyr also contributes to the observed differences between IL10 + CpG-stimulated MYC[low] and MYC[high] cells[20,45].

Furthermore, the context dependence of glutaminolysis was also reflected in the observation that oxidative phosphorylation of IL10 + CpG-stimulated MYC[low] cells was nearly independent from transaminase activity and Gln or fatty acids were not as dominantly involved as in MYC[high] cells. In contrast, MYC[high] cells are characterized by a strong dependence of oxidative phosphorylation from aminotransferases as observed by AOA treatment but also form fatty acids and Gln. The lower nitrogen

incorporation in Ala in IL10 + CpG stimulated MYC[low] cells as in MYC[high] cells, a stronger decrease of Ala amounts in MYC[high] cells after AOA treatment and the effect of the *GPT2* knockdown onto cell proliferation suggest that MYC[high] depend much more on anaplerosis to meet their increased energy, biosynthesis, and redox needs[46,47]. Whether the postulated Asp, ornithine, and arginine circuit is important as in M2-macrophages needs to be tested[48,49].

Pharmacological inhibition of aminotransferases has been suggested as a treatment strategy in MYC-overexpressing breast cancers[9]. Remembering that high aberrant *GOT2* is prognostic in DLBCL, transaminase inhibition might also represent an attractive treatment target in lymphoma. It is noteworthy that a subtype of DLBCL has been described as being dependent on cooperative STAT3 and NF-κB activation[26]. Furthermore, aberrant STAT3 and NF-κB activation, as well as aberrant *MYC* expression, are associated with inferior overall survival in DLBCL patients[50–54]. Although in some cases activation of these pathway may be explained by mutations inside the corresponding signaling cascades, a deregulated microenvironment might also be involved in the absence of complex intrinsic genetic lesions or high aberrant MYC.

Recently, the first metabolic distinction in DLBCL subtypes was described. OxPhos DLBCL cell lines were characterized by a strong mitochondrial component and a greater glycolytic flux when compared with other DLBCL cell lines[15]. Nevertheless, the precise network of factors regulating this or other metabolic landscapes is unknown. We now have strong evidence that STAT3/NF-κB activity driven by either a complex microenvironment in lymphomas or intrinsic genetic lesions is important in the metabolic reprogramming of cells different from OxPhos DLBCL. In addition, the recently described shifted OXPHOS-linked ATP synthesis in HL cells when compared with germinal center B-cells has to be complemented now by a strong glutaminolysis component[16]. We propose that Gln-derived Asp/nucleobases are important to HL cells, and that reductive carboxylation is more important than in other lymphoma cells.

Aminotransferases facilitate amino acid-mediated TCA anaplerosis and are emerging as critical determinants of oncogenesis. Recently, GOT1, GPT2, the branched-chain amino acid transferase 1, and phosphoserine aminotransferase 1 have been found to be essential for tumorigenesis of pancreatic adenocarcinoma, glioblastoma, and breast cancer, respectively, further supporting not only the herein described context-dependent role of GOT2 but also of GOT1 and GPT2[21,24,55,56].

Our findings not only provide functional validation of metabolic differences between different lymphoma subtypes. They also indicate that glutaminolysis and, in particular, GOT2 is a biomarker. Whether targeting GOT2 and glutaminolysis or inhibiting Jak/STAT and NF-κB signaling is more efficient in corresponding lymphoma subtypes needs to be investigated in prospective preclinical and clinical studies. However, as targeting MYC is still a challenge, targeting aminotransferases may prove to be an attractive treatment strategy or add-on therapy option in the future[9,57].

## Methods

**Cell culture**. P493-6 cells were a kind gift of Georg Bornkamm (Munich, Germany). TMD-8 and HBL-1 cells were gifted by Daniel Krappmann (Munich, Germany). All other cell lines (CA-46, BJAB, OCI-Ly3, KM-H2, and L-428) were obtained from the DSMZ Germany. All cell lines were identified and checked for cross contamination by Short Tandem Repeat (STR) profiling by the DSMZ Germany. Cell lines were routinely checked for Mycoplasma contamination by PCR using Venor-GeM Mycoplasma Detection Kit (Sigma-Aldrich, Munich, Germany). All cell lines were cultured in RPMI 1640 (Lonza, Cologne, Germany) containing 2 mM Gln supplemented with 10% Gibco[TM] FCS (Thermo Fisher Scientific, Waltham, MA, USA) and 1:100 penicillin/streptomycin (Thermo Fisher

Scientific) at 37 °C with 5% $CO_2$. To repress *MYC* in P493-6, the cells were treated with 1 ng/mL doxycycline for 24 h. Before the start of the stimulation experiments, the cell culture media was replaced by fresh media with or without adding 1 ng/mL doxycycline as indicated. The reduced expression of *MYC* by doxycycline remained reduced for up to 72 h.

For cell doubling experiments, P493-6 cells were stimulated every 24 h and cell number was counted 48 h after the first stimulation by using a hemocytometer and Trypan blue staining. Other cell lines were treated as indicated and cell count relative to the number of seeded cells was determined likewise after 48 h.

Stimulants, inhibitors, and metabolites used in this study are further described in the Supplementary Information. Concentrations of stimuli, inhibitors and metabolites used in this study are listed in Supplementary Table 2-4. For further stimulations of P493-6 cells with IL10, CpG, and IL10 + CpG, full working concentrations were used. It is noteworthy that after addition of metabolites, pH of cell culture media was adjusted to pH 7 by adding 1 M NaOH or 1 M HCl.

To stimulate P493-6 cells for transcriptomic and metabolic analysis in Supplementary Fig. 1, P493-6 cells were counted, centrifuged, and seeded in fresh cell culture medium at a density of $1 \cdot 10^6$ cells/mL. As depicted, cells were treated with either a single stimulus or combinations of at least two different stimuli, which could each take one of two different concentrations (see Supplementary Tables 2 and 5).

Small interfering RNAs (siRNAs) against *GOT1*, *GOT2*, *GPT2*, *STAT3*, *RELA* (all Smartpools, Dharmacon, Lafayette, USA) and scrambled control (Scrb, Invitrogen[TM] Silencer[TM]Select Negative Control No. 1, Thermo Fisher Scientific) were transfected into cells by electroporation using Amaxa® Nucleofector™ system (Lonza) according to the user manual (analog to protocol for Ramos cells). . siRNA sequences are listed in Supplementary Table 6. Cell line-specific modifications are shown Supplementary Table 7. P493-6 cell line was transfected in MYC[high] state and doxycycline was added 3 h after transfection to generate MYC[low] cells. All following experiments, including western blot analysis to analyze knockdown efficiency, were performed 24 h after knockdown.

**Real-time PCR analysis**. In all GE analyses, cells were collected and spiked with 1:10 *Drosophila* S2 cells as an external spike-in control[59]. RNA and complementary DNA was generated by NucleoSpin RNA kit (Machery-Nagel, Düren, Germany) and SuperScript II Reverse Transcriptase (Thermo Fisher Scientific), respectively. Following primer sequences were used: *GOT2* fwd: 5′-GAG AAC AGC GAA GTC TTG AAG AGT G-3′, *GOT2* rev: 5′-CTG GCT CCG ATC CTT AAG GCT-3′; *ACT42A* fwd: 5′-CTA AGC AGT AGT CGG GCT GG-3′, *ACT42* rev: 5′-GTC TGC AAT GGG TGT GTT CG-3′. Relative fold changes were calculated as $2^{-\Delta\Delta Ct}$ relative to *Drosophila* *ACT42A* and corresponding control cells[12].

**RNA sequencing**. One million P493-6 cells were spiked with 100,000 *Drosophila melanogaster* cells as external controls to normalize the GE data to the number of sample cells[59]. Total RNA was isolated using NucleoSpin Kit (Machery-Nagel). RNA sequencing (RNA-seq) libraries were prepared from 1 μg total RNA (containing human and *Drosophila* RNA). Library preparation was performed with the TruSeq RNA Sample Preparation Kit v2 (Illumina, San Diego, CA, USA). Libraries were sequenced in single end mode for 100 cycles on an Illumina HiSeq 2000.

We concatenated the genomes of Homo sapiens (GRCh38) and *D. melanogaster* (Ensembl BDGP5), and mapped all sequence libraries against this concatenated genome using TopHat version 2.0.13, indicating an unstranded sequencing protocol (library-type fr-unstranded) and default settings for the remaining parameters as described before[59].

Datasets were searched for ribosomal RNA reads by means of BioBloomtools (http://www.bcgsc.ca/platform/bioinfo/software/biobloomtools) using a score of 0.25 and kmer length 50, which were then removed from the bam files with custom code (between 1.5% and 3.4% of the reads). Next, RNA-seq read counts were assigned to Ensembl gene identifiers using featureCounts version 1.4.5 and Ensembl version 77 (50 samples), and featureCounts version 1.5.0 and Ensembl version 83 (15 samples) for the human genes. For the *Drosophila* gene annotation, Ensembl version 77 was used for both data sets.

From the data set of 50 samples with different stimulations and combinations thereof, transcripts per million (TPMs) were calculated for each gene with non-zero counts in at least one sample. If a gene had at least one zero count, a pseudo-count of one was added to all samples for this gene. The human gene TPMs were then normalized by Loess regression on the *Drosophila* TPMs and then served as input for the Lasso linear regression.

The data set consisting of 15 samples was normalized to constant numbers of sample cells by using size factors calculated by "DESeq2" from the *Drosophila* counts and applied to the human gene counts as described before[59]. We then carried out differential expression analysis using "DESeq2" with the design formula "CpG + IL10 + Myc + CpG:IL10." Genes with a count average smaller than "30" were discarded.

**Regression analysis**. Linear regression analysis of GE values from RNA-seq data (TPMs scaled to standard units) described in Supplementary Table 5 and metabolite measurements (scaled to standard units) was performed within the statistical framework R using Lasso regression within the 'glmnet' package. The Lasso could

select from coefficients for the individual stimuli and all pairwise combinations of external stimuli. The Lasso parameter lambda was selected such that it minimized the cross-validation error. In Supplementary Fig. 1, the Lasso regression coefficients are shown for the gene models with highest prediction accuracies ($R^2 > 0.8$, 9731 genes). Replicate samples per condition varied (see Supplementary Table 5 for details).

**Sample preparation for metabolic measurements.** Cells were counted 24 h after stimulation and $5 \times 10^6$ cells were centrifuged ($300 \times g$, 5 min, 4 °C). Supernatants were transferred into a new tube, whereas cell pellets were washed two times in cold phosphate-buffered saline (PBS) and resuspended in ice-cold 80% MeOH. Supernatants were filtered using 10 kDa Amicon® ultra centrifugal filters (Merck, Darmstadt, Germany) and centrifuged for 30 min, $4000 \times g$, at 4 °C. Metabolites from pellets were extracted using 80% methanol[64]. In brief, cell pellets were vortexed in 80% MeOH and centrifuged ($10,000 \times g$, 6 min, 4 °C). The supernatant was removed and the cell pellet was washed two more times with 80% MeOH. All supernatants were combined and dried for 2 h in a vacuum evaporator. Gln metabolism was investigated by stable isotope tracing. P493-6 MYC$^{high}$ and MYC$^{low}$ cells were washed once in warm PBS and seeded in Gln-free medium supplemented with 2 mM uniformly labeled $^{13}C_5$-Gln, 1-$^{13}C$-Gln, or α-$^{15}N$ Gln. MYC$^{low}$ cells were stimulated with IL10, CpG, and IL10 + CpG, respectively, as described before. Achievement of isotopic steady state was confirmed and no significant changes in isotope enrichment in the organic acids analysed were observed between 18 and 30 h of stimulation. Subsequently, all isotope tracing experiments were performed for 24 h.

**Metabolic analysis.** A 600 MHz Avance III spectrometer (BrukerBioSpin, Rheinstetten, Germany) was used to acquire one-dimensional (1D) 1H and two-dimensional (2D) 1H–13C heteronuclear single-quantum correlation spectra as published before[61]. Absolute concentrations from 1D and 2D nuclear magnetic resonance (NMR) data were calculated employing individual peak calibration factors and MetaboQuant[58]. Amino acids, tryptophan metabolites, and intermediates of the methionine and polyamine metabolism were determined by mass spectrometry (MS) as previously described[60,62,63]. For each metabolite, a stable isotope labelled internal standard was used.

Organic acids of the TCA cycle were analysed by high performance liquid chromatography (HPLC)–electrospray ionization (ESI)–tandem MS (MS/MS) employing an API 4000 QTrap mass spectrometer (ABSciex, Darmstadt, Germany) hyphenated to a 1200 SL HPLC (Agilent, Boeblingen, Germany). Negative mode ionization and multiple reaction monitoring with a transition for each isotopolog was performed. A Discovery HS F5-3 HPLC column (2.1 × 150 mm, 3 μm; Supelco, Bellefonte, PA, USA) with a Security Guard column (C18, Phenomenex, Aschaffenburg, Germany) and gradient elution with 0.1% formic acid in water (v/v) as mobile phase A and acetonitrile as mobile phase B was used.

For the measurement of amino acids and organic acids from stable isotope tracer experiments, the said methods were extended to cover the transitions of all expected isotopologs.

In addition to HPLC–MS/MS analysis, intracellular citrate was also analysed by gas chromatography (GC)–MS after derivatization. A 50 μL aliquot of the sample extract was dried followed by methoximation and silylation and subsequent GC–MS analysis employing the derivatization protocol and instrumental setup previously described[60]. An injection volume of 1 μL and splitless injection was used.

Nucleobases were analysed by HPLC–quad time of flight–MS after hydrolysis of nucleic acids. The pellet obtained after metabolite extraction described above was hydrolyzed with 200 μL 2 M HCl at 120 °C for 4 h. After hydrolysis, 1 M NaOH was used to neutralize the solution. The neutralized solution was then dried with a vacuum evaporator and the residue was re-solved in 100 μL of water for liquid chromatography–MS analysis. A Maxis QTOF-MS (Bruker Daltonics, Bremen, Germany) hyphenated to an Thermo Scientific Dionex Ultimate 3000 UHPLC system (Idstein, Germany) through an ESI source operated in positive ion mode was used. Separation was carried out on a Waters Atlantis T3 reversed-phase column (2.1 × 150 mm, 3 μm) with mobile phase A (0.1 % formic acid in $H_2O$, v/v) and B (0.1 % formic acid in acetonitrile, v/v) and gradient elution. The column was kept at 35 °C and a flow rate of 0.3 mL/min was employed.

MS data were acquired in a mass range of 50–1000 m/z with an acquisition rate of 2 spectra/s. Mass calibration was performed using sodium formate clusters (10 mM sodium formate in 50:50 (v/v) water/ isopropanol). For data analysis, the peak area of the $[M + H]^+$ peak of each possible isotopolog was integrated.

Correction for natural stable isotope abundance and tracer impurity in the tracing experiments was performed using an in-house tool to correct MS/MS data and IsoCor for full-MS data[65].

**GOT2 activity assay.** GOT2 activity was measured by an indirect colorimetrical and a direct NMR based method[66,67]. For colorimetrical analysis, 200 μL of sonificated cell lysate was incubated with 1 mL 2 mM α-KG/200 mM aspartate in 0.1 M phosphate buffer (pH 7.4) at 40 °C. After 60 min, 1 mL of the 2,4-dinitrophenylhydrazine (1 mM, solved in 1 N HCl) was added. After 20 min at room temperature (RT), 10 mL of 0.4 N sodium hydroxide were added. At the end of

exactly 30 min, the optical density of the solution was measured at 505 nm, using water as the blank. As controls 200 μL of each lysate without incubation at 40 °C was mixed with 2,4-dinitrophenylhydrazine. For calibration, a Pyr standard curve was used.

For NMR-based analysis, cells were sonificated in isotonic HEPES buffer (25 mM HEPES, 1 mM $NaH_2PO_4$, 106 mM NaCl$_2$, 19 mM KCl, 1 mM CaCl$_2$). Thirty microliters of cell lysates were mixed with 970 μL of 5 mM glutamate, 5 mM OAA, and 200 μM pyridoxal-5'-phosphate in isotonic HEPES buffer. Samples were incubated for indicated time points at 37 °C. Afterwards, enzyme activity was stopped by cooking at 99 °C for 7 min. The samples were frozen at – 80 °C, measured as described above and Asp and α-KG were quantified from 1D 1 H NMR spectra employing the Chenomx software suite (Chenomx, Inc., Edmonton, Canada).

**Analysis of respiration and glycolytic capacities.** OCR and ECAR rates were measured by means of an XF Analyzer (Seahorse Bioscience, North Billerica, MA, USA). Before seeding, cell culture plates were coated with Cell Tak (Corning, Inc., Corning, NY, USA) according to the manufacturer's instructions. Metabolites and inhibitors were added to ports and measurements were performed. P493-6 MYC$^{low}$ cells were stimulated for 24 h in medium supplemented with 11 mM Glc and/or 2 mM Gln. Unstimulated MYC$^{high}$ cells were grown in the same medium in parallel. The next day, cells were counted and 100,000 cells per well were seeded in XF Base medium (supplemented with 11 mM Glc and/or 2 mM Gln) according to manufacturer's protocol for suspension cells. After attachment to the surface cells were incubated for 30 min at 37 °C without the addition of $CO_2$ to equilibrate to measurement conditions. In the meantime corresponding inhibitors as indicated in the figures legends were prepared and loaded into the ports of a hydrated assay cartridge. Port loading and concentrations used are described in Supplementary Tables 8-10. Assay cartridge was calibrated, cell plate loaded into XF analyzer and cycles were performed as listed in Supplementary Table 11. From these data, basal parameters were calculated according to manufactures protocol (difference Port A to Port D). OCR dependency was calculated as followed:

$$\text{OCR dependency} \, (\%) = 100 - \left( \frac{\text{OCR}_{\text{Baseline}} - \text{OCR}_{\text{Inhibitor}}}{\text{OCR}_{\text{Baseline}}} \times 100 \right)$$

**Immunoblot analysis.** Western blot analysis was performed using 10% SDS polyacrylamide gels and polyvinylidene difluoride blotting membranes (Merck). To analyse knockdown efficiencies, the following antibodies were used: rabbit α-GOT1 (14886-1-AP, Proteintech, Rosemont, IL, USA), rabbit α-GOT2 (14800-1-AP, Proteintech), rabbit α-STAT3 (9132, Cell Signaling Technology, Danvers, MA, USA), rabbit α-p65 (4764, Cell Signaling Technology), and mouse α-GAPDH (ab8245, Abcam, Cambridge, UK). For visualization of the bound antibodies, secondary horseradish peroxidase-coupled antibodies from Santa Cruz Biotechnology (Dallas, TX, USA) were employed (sc-2004 and sc-2005). Uncropped versions of all blots can be found at the end of the Supplementary Information (Supplementary Fig. 13-19).

**Immunohistochemical analysis of GOT2.** A total of 137 B-cell lymphoma specimens were obtained from the files of the Kiel Lymph Node Registry according to ethical approval. All the lymphomas were classified according to the World Health Organization classification using standard histologic, immunohistochemical, and molecular criteria. All lymphomas were analyzed using tissue microarrays (TMAs) containing two cores of 1 mm or 0.6 mm (for the Molecular Mechanisms in Malignant Lymphoma Network specimens) for each case. Specimens were scored as negative (no staining of lymphoma cells), partially positive (< 50% of lymphoma cells), and positive (> 50% of lymphma cells are positive), respectively. Immunohistochemical staining was performed on TMAs using a rabbit α-GOT2 (14800-1-AP, Proteintech) at a dilution of 1:100 and using antigen retrieval citrate mbuffer pH 6.0. The corresponding secondary antibody was purchased from Medac (Wedel, Germany).

**Proteomic analysis.** The samples (three replicates, 50 μg each) were subjected to tryptic digestion using the GASP-protocol. Quantitative label-free proteomic analysis was accomplished by nano-HPLC-SWATH-MS. A TOP40 method was used for the library runs and variable SWATH-windows (60 windows from 400 to 1000 m/z) were used for SWATH-MS-runs. Data were normalized on cell count and statistical evaluation was done using the IBM SPSS Statistics 23 using analysis of variance and the Bonferroni post hoc test. Data were corrected for multiple testing according to Benjamini and Hochberg.

**Chromatin immunoprecipitation.** ChIP analysis of STAT3 and p65 was performed as described before[12]. Briefly, cells were stimulated with IL10, CpG, or IL10 + CpG for 1 h at 37 °C. Cells were washed with PBS and crosslinked with 2 mM disuccinimidyl glutarate in PBS for 45 min at RT. Secondary crosslinking was performed with 1% formaldehyde for 15 min at RT. The reaction was stopped by the addition of 0.1 M glycine for 10 min at RT. Cells were washed twice and lysed in 50 mM Tris-HCl pH 8.0, 2 mM EDTA, 1% Igepal, and 10% glycerol for 15 min

on ice. Nuclear pellets were resuspended in 50 mM Tris-HCl pH 8.1, 1% SDS, and 10 mM EDTA, and sonicated by means of a Bioruptor (Diagenode, Seraing, Belgium). One percent Triton X-100 was added to the sheared chromatin and debris were removed by centrifugation. For one IP, 50 µL of protein A-coupled magnetic beads (Life Technologies) were incubated with 5 µg of a-IgG (sc-2027, Santa Cruz Biotechnology), a-STAT3 (sc-482, Santa Cruz Biotechnology), or a-p65 (sc-372, Santa CruzBiotechnology) overnight at 4 °C. After washing, sheared chromatin was added overnight at 4 °C. Beads were washed with RIPA-buffer (50 mM HEPES-KOH pH7.6, 500 mM LiCl, 1 mM EDTA, 1% Igepal, and 0.7% sodium deoxycholate) and once with 1 M Tris-HCl pH 8.0, 0.5 M EDTA. Ten percent Chelex (Bio Rad Laboratories, Hercules, CA, USA) were added to the washed beads and heated for 10 min at 95 °C. Protein was digested by using Proteinase K (30 min, 65 °C) and afterwards inactivated by heating to 95 °C for 10 min. The *GOT2* locus was amplified by real-time PCR using the following primer pair: fwd: 5′-TCG CTG TGA CGT GGC TC-3′, rev: 5′-TGA AGG TAA GGA CAG GGA CTT C-3′.

**Statistical analysis**. Except for the Cox and regression models, statistical analysis was performed using GraphPad Prism 6. Standard normal distribution was assumed for all samples and equal variances were tested using the Brown–Forsythe test. Differences between groups were assumed to be statistically significant in all experiments for $p$-values < 0.05.

The Cox proportional hazards model was calculated using the R package 'survival' on the R-CHOP-treated DLBCL patients[18]. Activated B-cell like (ABC) and germinal center B-cell like (GCB) classified patients were included, while unclassified patients were excluded. In addition to the molecular DLBCL subtype, the IPI was included as a binary variable (1 if IPI > 2, 0 otherwise), age was included as a continuous variable, and *GOT2* expression was included as a binary variable (1 if *GOT2* expression > median, 0 otherwise).

**Data availability**. RNA sequence data that support the findings of this study have been deposited in the BioProject database of the NCBI with the accession numbers PRJNA312050 (https://www.ncbi.nlm.nih.gov/bioproject/?term=PRJNA312050;) and PRJNA356698 (https://www.ncbi.nlm.nih.gov/bioproject/?term=PRJNA356698). Proteomic data are available within the supplementary information as exel file. All other data of this study are available from the corresponding author upon reasonable request.

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

## Acknowledgements

We thank S. Hengst and C. Botz-von Drathen for excellent technical assistance. This work was supported by grants from the Federal Ministry of Education and Research (Bundesministerium für Bildung und Forschung, BMBF) within the network eBio "MMML-MYCSYS" and eMed MMML-Demonstrator (BMBF-FKZ 0316166E, 0316166 G, 031A428A, 031A428B) (D.K., R.S., J.C.E., W.G.). We thank all the members of the consortia "HaematoSys", "MMML-Demonstrators," and "MMML-Myc-Sys" for their assistance in preparing this manuscript.

## Author contributions

M.F., P.S., P.H., J.K., F.v.B, and X.S. performed most of the experimental work. P.S., P.H., K.D., J.R., and W.G. contributed to the specific analyses and data interpretation of metabolites and proteomics analysis. W.K. performed immunohistochemical analysis. J. D. performed respiratory analyses. F.T., P.P.-R., T.R., and J.C.E. performed the computational analysis of RNA sequencing data, regression, and survival analyses. J.C.E., W. G., R.S., L.T., and P.J.O. were involved in project design, data analysis, and final approval of the manuscript. M.F. and D.K. designed the research, analysed and interpreted data, and wrote the manuscript.

## Additional information

**Competing interests:** The authors declare no competing interests.

