## [Peer Review File · Nature Communications]

Reviewers' comments:

Reviewer #1 (Remarks to the Author):

The manuscript by Feist et al. follows up on their recent finding that the P493-6 MYC-dependent cell line could be stimulated to proliferate with IL-10 (via STAT) and CpG (via TLR 9-NFkB) independent of ectopic MYC, which could be turned off in the presence of tetracycline (or doxycycline). Here, they sought to document the metabolic differences between IL-10/CpG versus MYC dependent proliferation of P493-6 cells. While this study is restricted primarily to one model cell line (notwithstanding the inclusion of several DLBCL lines), the general principle of metabolic control by different pathways uncovered in this study is important. The authors document that while MYC-driven proliferation is dependent on glucose and glutamine flux, the IL-10/CpG program is less dependent on glutamine particularly for OXPHOS. Further, the inability of aspartate to rescue glutamine-deprived MYC-dependent cells but not IL-10/CpG dependent ones suggest an additional difference in metabolic wiring of these proliferating isogenic cells. The authors further document that STAT and NFkB pathways are downstream of IL-10/CpG through various markers and use of inhibitors, which do not seem to affect MYC-dependent cells. Both dependencies, however, appear to depend on the transaminase GOT2 which could be regulated by multiple transcription factors. These findings are important contributions to the literature; however, there are a number of technical issues that should be addressed.

1. The authors should provide immunoblot(s) to document to levels of MYC protein under various conditions used in the paper. At the minimum, please provide MYC protein levels for MYC high, MYC low control, MYC low plus IL-10, MYC low plus CpG, and MYC low plus IL-10/CpG combination. Further the authors should comment on how IL-10/CpG affect MYC high cells.
2. It would be highly instructive for the authors to provide light microscopic photographs of Wright-stained (or other equivalent stain) P493 (cytospined) cells under various conditions as well as cell size measurements (with flow cytometry or Coulter).
3. The experimental conditions for MYC low cells should be further clarify. The authors describe that MYC low cells were treated with 1nM tetracycline for 24h. Please clarify whether exposure to tetracycline is continuous with experiments extending more than 24 hours (eg, 13-carbon labeling studies).
4. For metabolomics studies, please clarify the normalization methods – total protein or cell number.
5. Figure 2C. The authors use m+5 isotopologue of citrate as evidence for reductive carboxylation. However, with a 24hour labeling, the extent of recycling through the TCA cycle cannot be definitely distinguished from reduction carboxylation unless special 13-carbon labeling of glutamine is used. In particular, use of 1-13carbon-labeled glutamine should confirm this. Through formal flux analysis, the extent of reductive carboxylation in P493 cells is low (Murphy et al. Metab Eng, 2013, 15:206-17). Hence, the authors should consider being circumspect with interpreting their data.
6. Page 9, line 202. Thymine appears to rescue MYChigh cells; statistics should be applied to the data before drawing the conclusion that 'adenine or thymine also proved unable...'
7. The authors should discuss the dependency of OCR in MYC-high cells on glutamine and yet CB-839 (glutaminase GLS inhibitor) influenced MYC-high cells similarly to MYC-low cells treated with IL-10/CpG.

Reviewer #2 (Remarks to the Author):

Feist et al report in their MS entitled "Transcriptional regulation of metabolic pathways by cooperative STAT/NF-κB signaling - aberrant GOT2 expression and glutamine addiction in lymphoma" that IL10/CpG stimulate B cell growth in a manner that depends n transcriptional upregulation of GOT2 and imply GOT2 may be a therapeutic target in lymphoma.

Some general questions.

- 1) What is the relative importance of transcriptional among other ednproduct and metabolic means of controlling GOT2 activity? In their system does GOT2 mRNAs correspond to activity?
- 2) What is the "MYCenvironment" (l. 45) and why is it relevant to the study on IL10/CpG?
- 3) Is GOT2 expressed in a tissue / lymphoma specific manner? This will be relevant to the expected toxicity of inhibitors.

Specifics:

- 1) The title is too long and almost incomprehensible. What is the key finding?
- 2) Changes in Fig. 1 are described as "synergistic" (l 106/7) – I guess the authors are saying that metabolic changes induced by IL10 and CpG are synergistic? What is the evidence for synergy? The panels in figure 1 seem to lack the baseline (MYC off and no treatment condition) and also the MYC on condition.
- 3) The labeling experiment is interesting. The key question seems to be whether this is this just a non-specific proliferative response or specific to the kind of stimulus? The authors point out similarities and differences between IL10/CpG and MYC, however I am puzzled about the implications of these results – what does it mean that there is more incorporation into ornithine etc..?
- 4) Figure 2 in part addresses this question with a series of nice rescue experiments showing a difference between IL10/CpG and MYC ocnditions in their requirements for OAA and pyruvate, respectively. Again, it remains unclear how the authors interpret these findings – I understand that its anabolic (l.218/9), but what dot he differences tell us?
- 5) These studies are somewhat limited by the use of a single cell line model (P493 cells).
- 6) The treatment studies and especially the rescue experiment to show relevance of the downstream pathways are nice. It would be nice if the authors would provide some additional background on compounds used (reference/in vivo use/additional drug targets).
- 7) Can the authors discuss the distinction between GOT1 and 2. The authors indicate that GOT2 is a "metabolic hub" in B cells. What other tissues express GOT2? Is the inhibitor selective for GOT1 vs 2?
- 8) The effects of GOT knockdown on proliferation in Fig 4B are not remarkable – is there a reason why this might be the case?

Reviewer #3 (Remarks to the Author):

The submitted paper highlights an important function of Jak/Stat signaling in regulating Got2 expression and glutamine/aspartate/nucleotide metabolism. Transcriptomic and metabolomic analysis of lymphoma cells with low MYC expression treated with various B-cell divered stimuli indicated that metabolic changes were significant. Tracer studies indicate increased glutaminolysis in stimulated cells, which supports proliferation. Regulation of the pathway by Jak/Stat is indicated, and correlation analysis in human clinical samples further supports a role for GOT2 expression in lymphoma progression.

This is a well presented study, though the impact seems more informative from a characterization standpoint as opposed to therapeutic (a credit to the authors for making the distinction). On the one hand, it is not surprising that metabolic gene expression is associated with proliferation. It is also clear (within the metabolic community) that aspartate is a critical nod for proliferation, and it is predominantly synthesized from Gln via GOT activity. In defense of this study, higher impact manuscripts (Nature) have been published showing GOT1 is important for cancer cell proliferation. I do see more novelty in deciphering the regulation of GOT2 expression by the Jak/Stat pathway, though this functionality may be Myc-dependent itself. With these caveats in mind, the study is well executed and data largely support the proposed concepts.

1. The central focus of the manuscript is (as stated by authors) on transaminase flux. Therefore

the more appropriate tracers are alpha-15N-glutamine. The authors will likely see ala and asp labeling as the same (since both are very low in media). Ser and gly are present in the media and therefore subject to dilution, which must be considered in any interpretation of data. This cuts to metabolic novelty, since the fact that gln usage increases with proliferation is well appreciated in the metabolic community. For example, the differential labeling of Asp and TCA intermediates (versus ala, gly, ser) is obvious from pathway architecture, as the other amino acids are glucose derived).

2. On a related note, alanine levels (which will be depleted) should be shown in the AOA treated cells, as this compound is not specific for GOT activity. Again, ser and gly are present in the media at much higher levels than asp and ala so they should be less impacted.

3. How are nucleotide levels changed in their stimulations and challenge conditions? These data would support function of GOT2 for nucleotide synthesis (rather than some other function).

4. Both Myc and Jak/Stat activation stimulate GOT2 expression, but are these regulatory functions truly independent? Can Jak/stat stimulate metabolism w/o increasing Myc levels? In looking at Myc vs Jak/stat dependence, tracer studies are recommended rather than relying on gene expression, as the functional output is important for demonstrating true regulation.

5. The data in Fig 4B is informative, but I am concerned about the low level of knockdown. Is Got1 really not important for Asp synthesis and proliferation here? With 30-40% expression remaining, there may not be a tangible effect on flux.

Minor

The authors should read (and cite) the manuscript by Murphy et al. (Met Eng 2013), which like the pub by Le et al. performed detailed 13C tracer analysis on the cell lines used here.

Line 202: why is thymine crossed out?

Response to Reviewers' comments:

Reviewer #1:

The manuscript by Feist et al. follows up on their recent finding that the P493-6 MYC-dependent cell line could be stimulated to proliferate with IL-10 (via STAT) and CpG (via TLR 9-NFkB) independent of ectopic MYC, which could be turned off in the presence of tetracycline (or doxycycline). Here, they sought to document the metabolic differences between IL-10/CpG versus MYC dependent proliferation of P493-6 cells. While this study is restricted primarily to one model cell line (notwithstanding the inclusion of several DLBCL lines), the general principle of metabolic control by different pathways uncovered in this study is important. The authors document that while MYC-driven proliferation is dependent on glucose and glutamine flux, the IL-10/CpG program is less dependent on glutamine particularly for OXPHOS. Further, the inability of aspartate to rescue glutamine-deprived MYC-dependent cells but not IL-10/CpG dependent ones suggest an additional difference in metabolic wiring of these proliferating isogenic cells. The authors further document that STAT and NFkB pathways are downstream of IL-10/CpG through various markers and use of inhibitors, which do not seem to affect MYC-dependent cells. Both dependencies, however, appear to depend on the transaminase GOT2 which could be regulated by multiple transcription factors. These findings are important contributions to the literature; however, there are a number of technical issues that should be addressed.

1. The authors should provide immunoblot(s) to document to levels of MYC protein under various conditions used in the paper. At the minimum, please provide MYC protein levels for MYC high, MYC low control, MYC low plus IL-10, MYC low plus CpG, and MYC low plus IL-10/CpG combination. Further the authors should comment on how IL-10/CpG affect MYC high cells.

Answer part A of the question: We performed corresponding analysis and added the requested data of immunoblots as a new **Suppl. Figure 9**.

First, we analysed the influence of IL10, CpG and IL10+CpG in P493-6Myclow cells at different time points (2, 4, 8 and 24 hours). IL10 alone increases the amount of Myc protein at very low levels, whereas stimulation by CpG alone was associated with a comparable increase in Myc protein levels as by IL10+CpG costimulation with a peak at 4 hours. At time point 24 hours the

Myc levels returned to the level of unstimulated cells (**Suppl. Figure 9A**). This is the first indication of the absence of a dominant effect of Myc onto cell proliferation under those conditions, as the level of Myc is comparable between CpG and IL10+CpG whereas the cell proliferation is completely different (**Suppl. Figure 9C**).

To test for the influence of this Myc protein increase onto the proliferation of P493-6 cells, we performed a Myc knockdown and measured cell proliferation (**Suppl. Figure 9B, C**). In P493-6^{MYC^{high}} cells the Myc protein amount is reduced by only about 2 fold, whereas the proliferation is strongly affected compared to stimulated P493-6^{MYC^{low}} cells. In IL10+CpG stimulated cells the „increase“ in Myc protein is reduced below the level of unstimulated cells, whereas the proliferation is still remarkably high and higher as from single stimuli. From this we concluded that there is no major function of Myc in IL10+CpG stimulated cell proliferation.

Within the main text the following paragraph was included on page 20 line 417:

.... To exclude that increased cell proliferation of IL10+CpG stimulated MYC^{low} cells was rather the result of increased MYC expression mediated by STAT3 and p65 activation the changes in Myc protein abundance as well as the effect of RNAi mediated Myc knockdown onto cell proliferation were monitored over 24 hours (**Suppl. Figure 9**). A transient low activation of Myc was observed after both CpG and IL10+CpG stimulation of MYC^{low} cells. However, cell proliferation was only increased upon IL10+CpG costimulation (**Suppl. Figure 9A**). Furthermore, the knockdown of Myc affected the proliferation of MYC^{high} cells much stronger despite a less pronounced reduction in Myc (**Suppl. Figure 9B, C**). Considering STAT3/p65 ChIP analysis, Jak/STAT and NF-κB inhibition and Myc knockdown data, we conclude that the synergistic effect of IL10 and CpG is driven mainly by STAT3 and NF-κB and not Myc. ...

Answer to part B of the question: The effect of IL10 and CpG onto Myc^{high} cells was not as intensively studied as with Myc^{low} cells. As MYC^{high} cells already have a very strong proliferation, only a slight increase in BrdU incorporation was observed. A corresponding graphic is shown below:

Legend to the figure: BrdU-incorporation in P493-6^{MYC^{high}} or P493-6^{MYC^{low}} cells after corresponding stimulations.

2. It would be highly instructive for the authors to provide light microscopic photographs of Wright-stained (or other equivalent stain) P493 (cytospined) cells under various conditions as well as cell size measurements (with flow cytometry or Coulter).

Answer: P493-6Myclow cells stimulated with IL10&CpG show increased cell size comparable to that what is observed when Myc^{low} cells are switched to Myc^{high} cells. A corresponding flow cytometric analysis is provided within **Suppl. Figure 3**. Wright staining of cytospined cells is not provided as the cells are not representative for their morphology after this centrifugation procedure.

Within the main text the following sentence was included on page 5 line 111:

... Interestingly, IL10+CpG stimulation of MYC^{low} cells resulted in an increase in cell size similar to that of MYC^{high} cells (**Suppl. Figure 2**). ...

and Legend to the Figure on page 39:

Supplemental Figure 2: P493-6 MYC^{low} cells show increased cell size upon IL10+CpG stimulation. (A) Forward/sideward scatter plots of flow cytometric measurements of unstimulated P493-6 MYC^{low}, IL10+CpG stimulated P493-6 MYC^{low} and P493-6 MYC^{high} cells respectively. (B) Graphical presentation of the relative cell size of unstimulated P493-6 MYC^{low}, IL10+CpG stimulated P493-6 MYC^{low} and P493-6 MYC^{high} cells

3. The experimental conditions for MYC low cells should be further clarify. The authors describe that MYC low cells were treated with 1nM tetracycline for 24h. Please clarify whether exposure to tetracycline is continuous with experiments extending more than 24 hours (eg, 13-carbon labelling studies).

Answer: Doxycycline was given 24 hours to reduce MYC expression before starting the experiment. To prepare cells for the corresponding analysis cells were pelleted, washed and transferred into new media containing freshly added doxycycline. The effect of doxycycline is still visible after 72 hours without adding it every 24 hours. A corresponding illustration is shown below.

Legend to the figure: Myc deprivation in P493-6^{MYC^{high}} cells. (A) Dose and time dependent reduction in Myc amounts after addition of doxycyclin. (B) Myc levels are still low after 72 hours of doxycyclin incubation. (C/D) Changes in BrdU labeling and cell viability over 72 hours after adding doxycyclin.

On page 28 line 624 within the section „Experimental Procedure“ the following sentences are added:

Before the start of the stimulation experiments the cell culture media was replaced by fresh media with or without adding 1ng/mL doxycycline as indicated. The reduced expression of MYC by doxycycline remained reduced for up to 72 hours.

- For metabolomics studies, please clarify the normalization methods – total protein or cell number.

Answer: In these experiments the metabolites were normalized based on cell numbers. Within the section „Experimental procedure“ the following paragraph was included (page 30, line 660):

..... and metabolite normalization was performed based on cell numbers.

5. Figure 2C. The authors use m+5 isotopologue of citrate as evidence for reductive carboxylation. However, with a 24hour labeling, the extent of recycling through the TCA cycle cannot be definitely distinguished from reduction carboxylation unless special 13-carbon labeling of glutamine is used. In particular, use of 1-13carbon-labeled glutamine should confirm this. Through formal flux analysis, the extent of reductive carboxylation in P493 cells is low (Murphy et al. Metab Eng, 2013, 15:206-17). Hence, the authors should consider being circumspect with interpreting their data.

Answer: We agree with the reviewers comment and performed corresponding experiments with 1-¹³C glutamine and observed a low reductive carboxylation of about 10% which seems to fit to the publication of Murphy et al.

The corresponding data are included into **Suppl. Figure 5** and the main text was changed in the following way on page 8 line 171:

.... To gain further insight into glutamine catabolism in IL10+CpG-stimulated MYC^{low} cells alpha-¹⁵N-glutamine tracing studies were performed (**Figure 2D**). The amino acids Ala, Asp, Glu and Pro showed increased enrichment of ¹⁵N in IL10+CpG-stimulated MYC^{low} cells. Even stronger was the incorporation in MYC^{high} cells. Importantly, labelled nitrogen derived from Asp and Gly was also observed in nucleobases (m+1, m+2) (**Figure 2E**). The very low level of nitrogen incorporation can be explained by the tracer, where only the nitrogen in alpha position of Gln was labelled to assess the contribution via Asp synthesis to nucleobase synthesis.

and Legend to the Figure 2/Suppl- Figure 5 on page10/ 40:

... Abundance of ¹⁵N isotopologues fractions of (**D**) amino acids and (**E**) nucleobases in unstimulated and IL10+CpG-stimulated P493-6^{MYC^{low}} as well as P493-6^{MYC^{high}} cells. ...

Supplemental Figure 5: Isotopic enrichment of citrate in unstimulated and IL10+CpG-stimulated P493-6^{MYC^{low}} and P493-6^{MYC^{high}} cells treated with 1-¹³C-glutamine for 24 hours. (A) Scheme of oxidative decarboxylation (blue) and reductive carboxylation (red). (B) Isotopic fractions of citrate in IL10+CpG-stimulated P493-6^{MYC^{low}} and P493-6^{MYC^{high}} cells. Mass spectra were analysed according to the different masses of citrate. (C) Isotopic enrichment of citrate in CA-46, OCI-Ly3 and L428 cells treated with ¹³C₅-glutamine for 24 hours. Isotopic fractions of citrate in corresponding lymphoma cell lines are presented. Mass spectra were analysed according to the different masses of citrate. Please refer also to Figure 2B for the scheme of oxidative decarboxylation.

In addition, within section „Experimental procedures“ and the Suppl. Methods file the analysis of isotope labelling experiments is now described in the following way on page 30 line 659:

.... For each metabolite a stable isotope labelled internal standard was used and metabolite normalization was performed based on cell numbers. For the measurement of amino acids and

organic acids from stable isotope tracer experiments, said methods were extended with additional mass windows to cover the transitions of all expected isotopologues. Correction for natural stable isotope abundance in the tracing data was performed using an in-house tool^{66–69}. Glutamine metabolism was investigated by stable isotopic tracing. P493-6 MYC^{high} and MYC^{low} cells were washed once in warm PBS and seeded in glutamine free medium supplemented with 2 mM ¹³C₅-glutamine, 1-¹³C-glutamine or alpha-¹⁵N glutamine. ...

Suppl. Methods file:

Instrumental analysis of samples from tracing experiments

Organic acids of the TCA cycle were analysed by HPLC-ESI-MS/MS employing an API 4000 QTrap mass spectrometer (ABSciex, Darmstadt, Germany) hyphenated to a 1200 SL HPLC (Agilent, Boeblingen, Germany). Negative mode ionization and multiple reaction monitoring (MRM) with a transition for each isotopologue was performed. A Discovery HS F5-3 HPLC column (2.1 × 150 mm, 3 µm; Supelco, Bellefonte, PA, USA) with a Security Guard column (C18, Phenomenex, Aschaffenburg, Germany) and gradient elution with 0.1% formic acid in water (v/v) as mobile phase A and acetonitrile as mobile phase B was used. Isotopologue analysis of amino acids was performed by HPLC-ESI-MS/MS analysis as previously described⁵. The method was adapted to monitor an MRM transition for each isotopologue.

In addition to HPLC-MS/MS analysis, intracellular citrate was also analysed by GC-MS after derivatization. A 50-µL aliquot of the sample extract was dried followed by methoximation and silylation and subsequent GC-MS analysis employing the derivatization protocol and instrumental setup previously described⁴. An injection volume of 1 µL and splitless injection was used.

Nucleobases were analysed by HPLC-QTOFMS after hydrolysis of nucleic acids. The pellet obtained after metabolite extraction described above was hydrolysed with 200 µL 2M HCL at 120°C for 4h. After hydrolysis, 1M NaOH was used to neutralize the solution. The neutralized solution was then dried with vacuum evaporator and the residue was re-solved in 100 µL of water for LC-MS analysis. A Maxis Impact QTOF-MS (Bruker Daltonics, Bremen, Germany) hyphenated to an Thermo Scientific Dionex Ultimate 3000 UHPLC system (Idstein, Germany) through an ESI source operated in positive ion mode was used. Separation was carried out on a Waters Atlantis T3 reversed-phase column (2.1×150 mm, 3 µm) with mobile phase A (0.1 % formic acid in H₂O, v/v) and B (0.1% formic acid in acetonitrile, v/v) and gradient elution. The column was kept at 35°C and a flow rate of 0.3 mL/min was employed.

MS-data were acquired in a mass range of 50–1000 m/z with an acquisition rate of 2 spectra/s. Mass calibration was performed using sodium formate clusters (10 mM sodium formate in 50:50 (v/v) water/isopropanol). For data analysis, the peak area of the [M+H]⁺ peak of each possible isotopologue was integrated.

Correction for natural stable isotope abundance and tracer impurity was performed using an in-house tool to correct MS/MS data and IsoCor for full-MS data⁶.

5. Dettmer, K. *et al.* Metabolite extraction from adherently growing mammalian cells for metabolomics studies: optimization of harvesting and extraction protocols. *Anal. Bioanal. Chem.* **399**, 1127–39 (2011).
6. van der Goot, A. T. *et al.* Delaying aging and the aging-associated decline in protein homeostasis by inhibition of tryptophan degradation. *Proc. Natl. Acad. Sci. U. S. A.* **109**, 14912–7 (2012).
7. Millard, P., Letisse, F., Sokol, S. & Portais, J.-C. IsoCor: correcting MS data in isotope labeling experiments. *Bioinformatics* **28**, 1294–1296 (2012).

6. Page 9, line 202. Thymine appears to rescue MYC^{high} cells; statistics should be applied to the data before drawing the conclusion that ‘adenine or thymine also proved unable...’

Answer: Data presented in the corresponding **Figure 2F/G** are analysed using one-way ANOVA estimations. Based on that no statistically significant increase in cell numbers was obtained upon addition of adenine or thymine, we conclude that both are unable to rescue cell proliferation in MYC^{high} cells.

7. The authors should discuss the dependency of OCR in MYC-high cells on glutamine and yet CB-839 (glutaminase GLS inhibitor) influenced MYC-high cells similarly to MYC-low cells treated with IL-10/CpG.

Answer: We thank the reviewer for this suggestion and present additional analyses included in **Suppl. Figure 8D** demonstrating the effects of Gln deprivation and AOA as well as CB-839 treatment.

- (I) In the absence of glutamine the oxidative phosphorylation in IL10+CpG stimulated cells is already higher in comparison to unstimulated MYC^{low} cells or MYC^{high} cell.
- (II) Adding Gln to Gln deprivation IL10+CpG stimulated MYC^{low} cells OCR changes are absent or low and also AOA of CD-839 are not changing much in OCR.

We analysed OCR in addition after application of inhibitors such as UK5099 to inhibit glycolysis, BPTES to inhibit glutaminolysis and etomoxir to interfere with FAO.

P493-6^{MYC^{high}} cells are significantly more dependent on Gln for respiration compared to IL10+CpG stimulated P493-6^{MYC^{low}} cells as proposed from the above described experiment. In addition, fatty acids are more important for MYC^{high} cells. Our interpretation of these findings is that IL10+CpG stimulated MYC^{low} cells use less glutamine and fatty acids for respiration compared with MYC^{high} cells. In future comprehensive analysis it has to be tested what are the substrates fueling the oxidative phosphorylation in IL10+CpG stimulated MYC^{low} cells in the absence of Gln.

The corresponding data are included into **Suppl. Figure 6** and the main text was changed in the following way on page 11 line 234:

... In addition, OCR was measured after respective inhibition of the mitochondrial pyruvate carrier (MPC1) by UK5099, carnitine palmitoyltransferase (CPT1) a key regulator of the beta-oxidation of long-chain fatty acids by etomoxir and glutaminolysis by the glutaminase inhibitor BPTES (**Suppl. Figure 6C**). As expected, BPTES affected MYC^{high} cells much more than IL10+CpG stimulated MYC^{low} cells. MYC^{low} and IL10+CpG stimulated MYC^{low} cells consumed less amounts of fatty acids than MYC^{high} cells. Furthermore, IL10+CpG stimulated cells are nearly independent in their oxidative phosphorylation from glucose.

Thus, our data suggest that IL10+CpG-stimulated B cells are characterized by glutaminolysis supporting anabolic functions and oxidative phosphorylation is supported in part by both fatty acids and glutamine. ...

and Legend to the Figure on page 41/43:

Supplemental Figure 6: (C) Basal OCR of IL10+CpG-stimulated MYC^{low} and MYC^{high} cells after inhibition of mitochondrial pyruvate carrier (MPC1) by UK5099, glutaminolysis affecting glutaminase by BPTES or fatty acid utilization inhibiting carnitine palmytoyltransferase (CPT1) by etomoxir.

Supplemental Figure 8:(D) OCR of IL10+CpG-stimulated MYC^{low} and MYC^{high} cells seeded in media without Gln. Gln, AOA, CB-839 and antimycin were added stepwise. One representative analysis out of three is shown. Error bars indicate technical variations.

Reviewer #2:

Feist et al report in their MS entitled “Transcriptional regulation of metabolic pathways by cooperative STAT/NF- κ B signaling - aberrant GOT2 expression and glutamine addiction in lymphoma” that IL10/CpG stimulate B cell growth in a manner that depends on transcriptional upregulation of GOT2 and imply GOT2 may be a therapeutic target in lymphoma.

- 1) What is the relative importance of transcriptional among other endproduct and metabolic means of controlling GOT2 activity? In their system does GOT2 mRNAs correspond to activity?

Answer: We thank the reviewer for this question about the complexity of regulatory levels within multilayer networks. The current approach in analysing global gene expression and developing predictive molecular signatures based on gene expression profiling is often based on a linear correlation of gene expression changes and a biological outcome. We are aware that the increase in gene expression does not automatically reflect higher protein amounts or enzyme activity. Therefore, we analysed the protein amount using a proteomics approach (SWATH). SWATH enables precise label-free quantification on proteome scale. Corresponding probes were analysed and showed that not only the gene expression but also the protein amount of for example GOT1 and GOT2 is increased after IL10+CpG stimulation in MYC^{low} cells.

The corresponding data are included into **Suppl. Table 1** and the main text was changed in the following way on page 5 line 111:

... Importantly, protein amounts were also increased (**Suppl. Table 1**).....

on page 15 line 308:

.... The transaminase GOT2 catalyzes the conversion of α -KG and Asp to Glu and OAA, while GOT1 catalyzes the reverse reaction in the malate-aspartate shuttle as long as the mitochondrial electron transport chain is functional^{20,22,23,24}. As shown in **Suppl. Table 1** the abundance of both enzymes GOT1 and GOT2 is increased after IL10+CpG stimulation of MYC^{low} cells. Thus the transcriptional activation is also translated into protein.

In addition within the Suppl. Methods file the SWATH-MS is described as follows on page 5:

Proteomic analysis using label-free quantitative LC-MS/MS (SWATH-MS)

The samples (three replicates, 50 µg each) were subjected to tryptic digestion using the GASP-protocol (10.1002/pmic.201400436). Quantitative label-free proteomic analysis was accomplished by nano-HPLC-SWATH-mass spectrometry as published previously (10.1021/acs.jproteome.6b00164), however, a TOP40 method was used for the library runs and variable SWATH-windows (60 windows from 400-1,000 m/z) were used for SWATH-MS-runs. Data were normalized on cell count and statistical evaluation was done using the IBM SPSS Statistics 23 using ANOVA and *post-hoc* t-tests. Data were corrected for multiple testing according to Benjamini and Hochberg⁹.

9. Hochberg, Y. & Benjamini, Y. More powerful procedures for multiple significance testing. *Stat. Med.* **9**, 811–8 (1990).

2) What is the “MYCroenvironment” (l. 45) and why is it relevant to the study on II10/CpG?

Answer: The play on words MYCroenvironment is an inspiration while performing this analysis as it was created in the mentioned editorial to discuss the recent observation of M. Yuneva and colleagues that even the „powerful“ oncogene Myc affects cell metabolism in a context dependent manner which includes factors of the tumor microenvironment.

Nevertheless we changed this paragraph in the following way on page 3 line 46:

... There is growing evidence for a cooperative role of stromal factors in the shift from quiescence to proliferation and the transformation of lymphocytes¹⁻³. A fundamental question is how microenvironmental factors and the intracellular processes activated by them regulate metabolic reprogramming. A deeper understanding of the differences between processes activated by either stromal or well-known single or complex genetic drivers such as the c-MYC (MYC) are crucial to the development of novel targeted cancer therapies.

MYC has been shown to increase glutaminolysis and glycolysis to support tumour cell proliferation⁴. A central role of glutaminolysis in resting and proliferating B cells was shown for the MYC-inducible human B cell line P493-6, which serves as a model of MYC-driven lymphoma or normal B cells⁵. In this model, MYC drives glutamine catabolism via the aberrant expression of glutaminase (GLS). Up-regulation of GLS increases the conversion of glutamine (Gln) to glutamate (Glu), which can be further metabolized to alpha-ketoglutarate (α-KG)⁶⁻⁹. The latter is a key metabolite involved in energy production via the tricarboxylic acid cycle (TCA), and it is also used as a biosynthetic precursor. Notably, α-KG may be synthesized from Glu by two different mechanisms: deamination by glutamate dehydrogenases (GLUD1/2) or transamination by aspartate transaminases (GOT1/2) or alanine transaminases (GPT1/2). ...

3) Is GOT2 expressed in a tissue / lymphoma specific manner? This will be relevant to the expected toxicity of inhibitors.

Answer: Based on additional immunohistochemistry analysis GOT2 is expressed in a tissue and lymphoma specific manner. We showed that there is aberrant GOT2 in HL, BL and DLBCL The corresponding data are included into **Figure 7** and new **Table 1** and the main text was changed in the following way on page 22 line 457:

... Next GOT2 was analysed using immunohistochemistry (**Table 1, Figure 7B**). We observed that normal lymphocytes are usually not stained with anti-GOT2 antibodies with the exception for some cells within the germinal centre (**Figure 7B**, tonsil). Lymphoma tissues of HLs, DLBCLs and BLs patients, in contrast, stained strongly for GOT2 thus supporting gene expression data (**Figure 7A,B**).

Table 1: Immunohistochemical analysis of GOT2 distribution in B cell lymphoma. Partially positive (<50% of lymphoma cells are stained with α -GOT2 antibody) and positive (>50% of lymphoma cells are positive for GOT2). For additional details see also Figure 7B and Suppl. Material and Methods section.

	Number of cases	Number of cases	
		positive	partial positive
Hodgkin lymphoma	15	15	0
Diffuse large B cell lymphoma	10	8	2
Burkitt lymphoma	6	6	0
Mediastinal B cell lymphoma	4	4	0
Acute B cell leukemia	5	2	3

In addition within the Suppl. Methods file the IHC is described in the following way on page 5:

Immunohistochemical analysis of GOT2 protein in tissue sections from lymphoma patients

A total of 137 B-cell lymphoma specimens were obtained from the files of the Kiel Lymph Node Registry according to ethical approval. All the lymphomas were classified according to the World Health Organization classification using standard histologic, immunohistochemical, and molecular criteria. All lymphomas were analyzed using TMAs containing two cores of 1 mm or 0.6 mm (for the Molecular Mechanisms in Malignant Lymphoma Network specimen) for each case. Scoring was performed according to the recent publication distinguishing negative (no staining of lymphoma cells), partially positive (<50% of lymphoma cells) and positive (>50% of lymphoma cells are positive). Immunohistochemical staining was performed on TMAs using a rabbit α -GOT2 antibody (#14800-1-AP, Proteintech), at 1:100 and using antigen retrieval citrate-buffer pH6,3. The corresponding secondary antibody was purchased from Histofine/medac.

Specifics:

- 1) The title is too long and almost incomprehensible. What is the key finding?

Answer: Although agreeing with the reviewer that the title might be too long, we would like to point out that the key finding is about a combined activity of Jak/STAT and NF- κ B signaling as important for the GOT2 associated glutamine addiction in aggressive lymphomas. We changed the title in the following way:

Cooperative STAT/NF- κ B signalling and metabolic reprogramming - aberrant GOT2 expression and glutamine addiction in lymphoma.

- 2) Changes in Fig. 1 are described as “synergistic” (1 106/7) – I guess the authors are saying that metabolic changes induced by IL10 and CpG are synergistic? What is the evidence for synergy? The panels in figure 1 seem to lack the baseline (MYC off and no treatment condition) and also the MYC on condition. The panels in figure 1 seem to lack the baseline (MYC off and no treatment condition) and also the MYC on condition.

Answer: We thank the reviewer for this comment. Based on the linear regression analysis summarized within the **Suppl. Figure 1** both global gene expression and analysed metabolites are changed in a synergistic way after IL10+CpG stimulation of P493-6 MYC^{low} cells. The corresponding heatmaps represent the regression coefficient not the Log2FC.

In **Figure 1** are changes on gene expression and metabolite abundance depicted as Log2FC. White color indicates that a given stimulus such as for example IL10 causes no change on a specific gene or metabolite compared to unstimulated MYC^{low} cells. The Myc-off and no treatment conditions in **Figure 1** are already included within the comparisons and thus not separately shown. The heatmaps are built to show the differential amounts of metabolites or gene expression changes in relation to unstimulated probes.

We modified the legend for **Figure 1** and the **Suppl. Figure 1** in the following way:

Figure 1: Activation of cell metabolism and global gene expression by combined stimulation of P493-6 cells with IL10 and CpG

(A) Heat map of changes in intracellular metabolites depicted as Log2FC after IL10 and/or CpG stimulation of P493-6^{MYC^{low}} cells in relation to unstimulated cells. (B/C) Heat map of gene expression changes associated with glycolysis (GO-term) (B) and glutaminolysis (GO-term) (C) after IL10 and/or CpG stimulation of P493-6^{MYC^{low}} cells presented as Log2FC. (D/E) Relative cell number of IL10+CpG-stimulated P493-6^{MYC^{low}} cells (D) and unstimulated P493-6^{MYC^{high}} cells (E) grown in media with and without either glucose (-Glc) or glutamine (-Gln). Mean and SD of three independent experiments are given. For additional details to estimate the synergy of IL10 and CpG calculated by linear regression see also Suppl. Figure 1.

Supplemental Figure 1: IL10+CpG stimulation of P493-6 cells synergistically induces gene expression and metabolic changes. P493-6 cells were stimulated as described before¹³. The effects of stimuli and stimuli combinations on global gene expression (A) and intracellular metabolite abundance (B) were calculated by linear regression. The regression coefficients are given in the heatmaps. It is obvious that all stimuli were capable to increase gene expression of the majority of genes, while nearly no genes were downregulated. Administration of all possible pairwise combinations of stimuli induced synergistic

effects on gene expression and metabolite abundance, evident from the interaction terms of the linear model (e.g. the coefficient CpG:IL10 representing the additional change in gene expression or metabolite abundance beyond the sum of the coefficients of IL10 and CpG). In the case of strong positive synergy the combinations of stimuli resulted in a larger increase in gene expression/metabolite amount than the simple addition of the single stimulation effects. Strong positive synergistic effects on gene expression and metabolites are shown in red or negative in blue.

- 3) The labeling experiment is interesting. The key question seems to be whether this is this just a non-specific proliferative response or specific to the kind of stimulus? The authors point out similarities and differences between IL10/CpG and MYC, however I am puzzled about the implications of these results – what does it mean that there is more incorporation into ornithine etc..?

Answer: Indeed one key question in all analysis about cell proliferation and metabolism is about the impact of metabolic changes for cell cycle progression as a prerequisite for cell doubling. To our understanding the combined activity of the Jak/STAT and NF- κ B pathways is affecting both, activating critical G1/S cell cycle phase regulating genes as CDK4/6 and metabolic genes supporting the biosynthesis processes important for a successful S-Phase (Feist et al. Int J Cancer 2017; Hermeking et al. PNAS 2000, Zachary et al. Cancer Discovery 2015). Although, still a majority of cell cycle researchers are stuck to classical cell cycle check point models, there is compelling evidence that the metabolism is tightly linked to cell cycle progression involving for example PFKFB3 and GLS1 (Colombo SL et al. PNAS 2010; Moncada S et al. Biochem J 2012).

The difference between Myc and Jak/STAT and NF- κ B pathways in order to regulate gene expression are mostly quantitative rather than qualitative. In addition, based on our observations we are developing the hypothesis that the way how glutaminolysis is involved in anabolic and catabolic processes is different between stimulated MYC^{high} and MYC^{low} cells or corresponding cell lines with aberrant Myc or STAT/NF- κ B. The rescue experiments and the measurement of the OCR revealed that STAT/NF- κ B regulated processes are supporting anabolic processes to provide aspartate and nucleobase whereas aKG is more important in MYC^{high} cells. Furthermore, the oxidative phosphorylation is less dependent from glutamine and fatty acids in IL10+CpG stimulated MYC^{low} cells when compared to MYC^{high} cells. Also intriguing is the observation about the differences in alanine production from glutamine or its accumulation after AOA treatment as revealed by additional experiments presented in **Figure 3C**. The incorporation of labelled atoms into ornithine is likely, as ornithine is directly derived from glutamate and proline and therefore not surprising. Hence, we removed the detailed description of ornithine from the manuscript.

- 4) Figure 2 in part addresses this question with a series of nice rescue experiments showing a difference between IL10/CpG and MYC conditions in their requirements for OAA and pyruvate, respectively. Again, it remains unclear how the authors interpret these findings

– I understand that its anabolic (1.218/9), but what do the differences tell us?

Answer: Data presented in **Figure 2** show (i) that glutamine is metabolized in both Myc^{high} and IL10+CpG stimulated MYC^{low} cells in a comparable way. Nevertheless the fractional abundance of m+4 citrate indicated that Myc^{high} cells might use glutamine in a more prominent way for oxydative phosphorylation. Furthermore, rescue experiments showed that aspartate and nucleobases are able to supplement for glutamine deficiency in IL10+CpG stimulated MYC^{low} cells, whereas α -KG is doing so in MYC^{high} cells.

We performed additional analysis to measure OCR dependency. We used inhibitors UK5099 to inhibit glycolysis, BPTES to inhibit glutaminolysis and etomoxir to interfere with fatty acid oxydation.

As shown in the new **Suppl. Figure 6** oxydative phosphorylation in MYC^{high} cells depends significantly stronger on Gln compared to IL10+CpG stimulated MYC^{low} cells.

Our interpretation of these findings is that Gln is less important for IL10+CpG stimulated MYC^{low} cells.

For a better understanding, we added more explanations in the following paragraph on page 11 line 234:

... In addition, OCR was measured after respective inhibition of the mitochondrial pyruvate carrier (MPC1) by UK5099, carnitine palmitoyltransferase (CPT1) a key regulator of the beta-oxidation of long-chain fatty acids by etomoxir and glutaminolysis by the glutaminase inhibitor BPTES (**Suppl. Figure 6C**). As expected, BPTES affected MYC^{high} cells much more than IL10+CpG stimulated MYC^{low} cells. MYC^{low} and IL10+CpG stimulated MYC^{low} cells consumed less amounts of fatty acids than MYC^{high} cells. Furthermore, IL10+CpG stimulated cells are nearly independent in their oxidative phosphorylation from glucose.

Thus, our data suggest that IL10+CpG-stimulated B cells are characterized by glutaminolysis supporting anabolic functions and oxidative phosphorylation is supported in part by both fatty acids and glutamine.

5) These studies are somewhat limited by the use of a single cell line model (P493 cells).

Answer: We thank the reviewer for these suggestions and performed additional analyses of glutamine and transaminase dependence on cell proliferation in the additional cell lines as KM-H2, BJAB, TMD8 and HBL-1. Furthermore, tracing experiments with uniformly labelled ¹³C-glutamine were performed for L428, CA-46 and OCI-Ly3. These data are included into **Suppl. Figure 7** and **Suppl. Figure 5C**.

The main text is changed in the following way on pages 12 line 259:

... Treatment of cells with CB-839 or AOA for 24 hours caused a strong reduction in cell proliferation in both IL10+CpG-stimulated MYC^{low} and unstimulated MYC^{high} cells (**Figure 3A**). The observed decrease was comparable to that caused by Gln starvation. Similar effects were

also observed for lymphoma cell lines representative of constitutive Jak/STAT and NF- κ B pathway activation (OCI-Ly3, L428) or aberrant MYC activity (CA46) (**Figure 3B**). The similar effects of Gln deprivation and AOA treatment were also demonstrated for the additional B cell lines BJAB, TMD-8, HBL-1, and KM-H2 thus underscoring the importance of transaminases in sustaining B cell lymphoma proliferation (**Suppl. Figure 7**). ...

on pages 14 line 289

.... Using $^{13}\text{C}_5$ -glutamine glutaminolysis was further characterized in the OCI-Ly3, L428 and CA-46 lymphoma cell lines. Cells were labelled with $^{13}\text{C}_5$ -glutamine for 24 hours and the fractional abundance of ^{13}C isotopologues of citrate was determined (**Suppl. Figure 5C**). As for the P493-6 cells, the m+4 isotopologue was the most abundant in CA-46 and OCI-Ly3 cells. In L428 cells, in contrast, the m+5 isotopologue dominated.

Supplemental Figure 5: Isotopic enrichment of citrate in unstimulated and IL10+CpG-stimulated P493-6^{MYClow} and P493-6^{MYChigh} cells treated with 1- ^{13}C -glutamine for 24 hours. (A) Scheme of oxidative decarboxylation (blue) and reductive carboxylation (red). (B) Isotopic fractions of citrate in IL10+CpG-stimulated P493-6^{MYClow} and P493-6^{MYChigh} cells. Mass spectra were analysed according to the different masses of citrate. (C) Isotopic enrichment of citrate in CA-46, OCI-Ly3 and L428 cells treated with $^{13}\text{C}_5$ -glutamine for 24 hours. Isotopic fractions of citrate in corresponding lymphoma cell lines are presented. Mass spectra were analysed according to the different masses of citrate. Please refer also to Figure 2B for the scheme of oxidative decarboxylation.

Supplemental Figure 7: NF- κ B and STAT3 activation in different lymphoma cell lines and the effect of Gln deprivation or AOA treatment on lymphoma cell line proliferation. (A) Immunoblot analysis of BJAB, CA-46, OCI-Ly3 and TMD8 cells for STAT3 and p65 phosphorylation, and I κ B α protein amounts with and without ACHP or Ruxolitinib treatment as described in **Figure 6**. (B) Changes in relative cell numbers of B cell lines BJAB, KM-H2, HBL1, TMD8 after glutamine deprivation and AOA treatment are shown.

- 6) The treatment studies and especially the rescue experiment to show relevance of the downstream pathways are nice. It would be nice if the authors would provide some additional background on compounds used (reference/in vivo use/additional drug targets).

Answer: Within our analysis we used chemical inhibitors to affect glutaminolysis (CB-839, AOA) and inhibitors targeting Januskinases and Inhibitor of I κ B-kinases (Ruxolitinib, ACHP). From these inhibitors AOA has been studied as a treatment for tinnitus, whereas CB-839 is still in clinical trials. Ruxolitinib is approved to treat Polycythaemia vera or Myelofibrosis.

- 7) Can the authors discuss the distinction between GOT1 and 2. The authors indicate that GOT2 is a “metabolic hub” in B cells. What other tissues express GOT2? Is the inhibitor selective for GOT1 vs 2?

Answer: GOT1 is a cytoplasmic aminotransferase, whereas GOT2 is localized within the mitochondrion. Both enzymes are important within the aspartate-malate shuttle. We performed additional IHC analysis.

The conclusion that GOT2 is a metabolic hub is based on both the *in vitro* analysis of cell lines and the observation of aberrant GOT2 expression and protein levels in patients samples but also that the GOT2 gene expression is predictive in DLBCL patients.

Although important in glutaminolysis, the gene expression of for example GLS/GOT1/GLUD is not increased comparing normal B cells and lymphoma cells, whereas GOT2 is aberrantly expressed in aggressive lymphoma.

In our IHC analysis we were using liver samples to establish the GOT2 staining for lymphoma samples. In reactive lymphatic tissues as tonsil only a selected number of lymphocytes is positive for GOT2.

AOA is an inhibitor of transaminase and therefore not very specific for only GOT1/GOT2. That was the reason to perform additional RNAi based knockdowns of additional transaminases. We decided to test for GPT2 (glutamic pyruvate transaminase).

A decreased GPT2 amount resulted in the reduction of cell proliferation of MYC^{high} cells. This is interesting as it is specific for MYC^{high} cells. This supports to our understanding the data on changes of Ala after AOA treatment.

However, a final answer to the question can be given perhaps only after a complete knockout of corresponding genes but current strategies for CRISPR/Cas mediated knockdown are not always effective in transformed B cells or associated with strong off-target effects.

The main text is changed in the following way on pages 15 line 326:

.... Differences between the effects of AOA treatment and GOT2 knockdown indicated the involvement of other transaminases. This prompted us to investigate the glutamate pyruvate transaminase (GPT2), the gene expression of which had increased less in IL10+CpG stimulated MYC^{low} cells as in MYC^{high} cells. No significant decrease in cell proliferation was observed after GPT2 knockdown in IL10+CpG stimulated MYC^{low} cells. In contrast cell proliferation of MYC^{high} cells was decreased after GPT2 knockdown (**Figure 4E/F**). A specific role for GPT2 in MYC^{high} cells is in line with the above described higher nitrogen incorporation into Ala and the decrease in Ala abundance after AOA treatment and the recent description of cells with cooperating oncogenic mutations ²⁵. ...

On page 24 line 505 and page 26 line 552 within the discussion also the following paragraphs were included:

.... Thus, STAT3/p65-dependent upregulation of GOT2 is a new important mechanism in the mediation of glutaminolysis during lymphocyte proliferation and transformation **albeit additional**

aminotransferases might be involved in the used B cell model cell line.....

..... The different glutaminolysis networks identified herein, were IL10+CpG stimulated MYC^{low} cells use less Gln for oxidative phosphorylation compared to MYC^{high} cells fits well with previous data demonstrating that MYC-dependent osteosarcoma cells use glutamine for cellular respiration ¹⁰. Therefore, we propose that GOT2 serves as a metabolic hub, providing cells with Asp and/or α -KG for energy demands and biosynthetic processes such as nucleotide biosynthesis in a context-dependent manner. Further, transaminases other than GOT2 such as GPT2 may contribute to the observed differences. GPT2 promotes Gln-dependent anaplerosis by catalyzing the reversible addition of an amino group from glutamate to pyruvate generating Ala and α -KG to fuel the TCA. It is believed that this provides a link between glycolysis, glutaminolysis and TCA anaplerosis. Smith et al. provided evidence for a model in which GPT2 is critical for the cancer phenotype including glutamine-driven TCA anaplerosis and coupling of glycolytic pyruvate output to glutamine catabolism ²⁵. Interestingly, GPT2 was dispensable in not fully transformed cells similar as observed herein for IL10+CpG stimulated MYC^{low} cells. Thus it seems to be reasonable that the Warburg effect supports oncogenesis via GPT2-mediated coupling of pyruvate production to glutamine catabolism in MYC^{high} cells but not IL10+CpG stimulated MYC^{low} cells ²⁵. It remains to be seen whether the observed oxidative decarboxylation of malate into pyruvate also contributes to the observed differences between IL10+CpG stimulated MYC^{low} and MYC^{high} cells ^{21,49}.

8) The effects of GOT knockdown on proliferation in Fig 4B are not remarkable – is there a reason why this might be the case?

Answer: We agree with the reviewer's comment. One explanation for this could be that the knockdown of the GOTs is not reaching a threshold important for a stronger functional outcome. Another possibility is that other aminotransferases are involved in this cell model and that GOT2 in our cell systems is „only“ one amongst additional transaminases in glutaminolysis. The use of AOA shows a strong effect onto IL10+CpG mediated MYC^{low} cell proliferation. The above described analysis with GPT2 supports this view.

Reviewer #3:

The submitted paper highlights an important function of Jak/Stat signaling in regulating Got2 expression and glutamine/aspartate/nucleotide metabolism. Transcriptomic and metabolomic analysis of lymphoma cells with low MYC expression treated with various B-cell divered stimuli indicated that metabolic changes were significant. Tracer studies indicate increased glutaminolysis in stimulated cells, which supports proliferation. Regulation of the pathway by Jak/Stat is indicated, and correlation analysis in human clinical samples further supports a role for GOT2 expression in lymphoma progression.

This is a well presented study, though the impact seems more informative from a characterization standpoint as opposed to therapeutic (a credit to the authors for making the distinction). On the one hand, it is not surprising that metabolic gene expression is associated with proliferation. It is also clear (within the metabolic community) that aspartate is a critical nod for proliferation, and it is predominantly synthesized from Gln via GOT activity. In defense of this study, higher impact manuscripts (Nature) have been published showing GOT1 is important for cancer cell proliferation. I do see more novelty in deciphering the regulation of GOT2 expression by the Jak/Stat pathway, though this functionality may be Myc-dependent itself. With these caveats in mind, the study is well executed and data largely support the proposed concepts.

1. The central focus of the manuscript is (as stated by authors) on transaminase flux. Therefore the more appropriate tracers are alpha-15N-glutamine. The authors will likely see ala and asp labeling as the same (since both are very low in media). Ser and gly are present in the media and therefore subject to dilution, which must be considered in any interpretation of data. This cuts to metabolic novelty, since the fact that gln usage increases with proliferation is well appreciated in the metabolic community. For example, the differential labeling of Asp and TCA intermediates (versus ala, gly, ser) is obvious from pathway architecture, as the other amino acids are glucose derived).

Answer: We agree with the reviewer's recommendation. Corresponding alpha-¹⁵N-glutamine tracings analysis were performed. Amino acids and nucleobases were measured. The corresponding data are included into **Figure 2** and the main text was changed in the following way on page 8 line 171:

.... To gain further insight into glutamine catabolism in IL10+CpG-stimulated MYC^{low} cells alpha-¹⁵N-glutamine tracing studies were performed (**Figure 2D**). The amino acids Ala, Asp, Glu and Pro showed increased enrichment of ¹⁵N in IL10+CpG-stimulated MYC^{low} cells. Even stronger was the incorporation in MYC^{high} cells. Importantly, labelled nitrogen derived from Asp and Gly was also observed in nucleobases (m+1, m+2) (**Figure 2E**). The very low level of nitrogen incorporation can be explained by the tracer, where only the nitrogen in alpha position of Gln was labelled to assess the contribution via Asp synthesis to nucleobase synthesis. ...

Figure 2: Proliferation of IL10+CpG-stimulated and MYC overexpressing cells is dependent on different glutamine-derived metabolites.

(A) Isotopic enrichment of metabolites in unstimulated and IL10+CpG-stimulated P493-6^{MYC^{low}} and P493-6^{MYC^{high}} cells treated with ¹³C₅-glutamine. (B) Scheme of oxidative decarboxylation (blue) and reductive carboxylation (red). Both can be distinguished by isotopic labelling in mass spectrometry. (C) Isotopic fractions of citrate in IL10+CpG-stimulated P493-6^{MYC^{low}} and P493-6^{MYC^{high}} cells. Mass spectra were analysed according to the different masses of citrate. Data on 1-¹³C-glutamine labelling are summarized in Suppl. Figure 5. Abundance of ¹⁵N isotopologues fractions of (D) amino acids and (E) nucleobases in unstimulated and IL10+CpG-stimulated P493-6^{MYC^{low}} as well as P493-6^{MYC^{high}} cells. (F+G) Cell numbers of IL10+CpG-stimulated P493-6^{MYC^{low}} and P493-6^{MYC^{high}} cells grown in Gln deprived and metabolite treated media. Cells were either treated with TCA metabolites (Pyr = pyruvate, α-KG = dimethyl 2-oxoglutarate, Mal = diethyl malate, OAA = oxaloacetate) in (F) or aspartate (Asp) and nucleobases (A =adenine, T = thymine) in (G) for 24 hours. In all experiments, mean±SD of at least three independent experiments and one-way ANOVA results are given (*p<0.05, **p<0.01, ***p<0.001).

2. On a related note, alanine levels (which will be depleted) should be shown in the AOA treated cells, as this compound is not specific for GOT activity. Again, ser and gly are present in the media at much higher levels than asp and ala so they should be less impacted.

Answer: We thank the reviewer for this suggestion. We measured correspondingly Ala with/without AOA. Interestingly, the amount of Ala dropped only in MYC^{high} cells significantly. Also taking into account that in IL10+CpG treated MYC^{low} cells nitrogen is incorporated at lower amounts into Ala and corresponding OCR data, it can be concluded that glutamine is used in MYC^{high} cells more for energy supply compared to IL10+CpG stimulated MYC^{low} cells.

The new data are now part of **Figure 3C** and the main text was extended in the following way on page 13 line 277:

.... Importantly, measurements of intracellular amino acid abundance revealed that Asp levels dropped significantly after AOA treatment in both IL10+CpG-stimulated MYC^{low} and MYC^{high} cells (**Figure 3C**). The intracellular levels of other amino acids such as Glu, Gly, Pro, or Ser were only slightly affected by transaminase inhibition. Of note is that the amount of Ala was significantly affected after AOA treatment only in MYC^{high} cells (**Figure 3C**).

Figure 3: Proliferation of P493-6 and B-cell lymphoma cells requires transaminase activity.

(A) Inhibition of cell doubling of IL10+CpG-stimulated P493-6^{MYC^{low}} and P493-6^{MYC^{high}} P493-6 cells after 24h incubation with CB-839 or AOA. (B) Inhibition of cell doubling of the B-cell lymphoma cell lines CA-46, OCI-Ly3 and L428 using CB-839 or AOA for 24h. (C) Intracellular amino acid abundance of IL10+CpG-stimulated MYC^{low} and MYC^{high} cells 24h after AOA treatment and stimulation. Percentages of DMSO-treated cells are displayed for each amino acid. Means±SD of three independent experiments are given.

on page 11 line 234:

In addition, OCR was measured after respective inhibition of the mitochondrial pyruvate carrier (MPC1) by UK5099, carnitine palmitoyltransferase (CPT1) a key regulator of the beta-oxidation of long-chain fatty acids by etomoxir and glutaminolysis by the glutaminase inhibitor BPTES (**Suppl. Figure 6C**). As expected, BPTES affected MYC^{high} cells much more than IL10+CpG stimulated MYC^{low} cells. MYC^{low} and IL10+CpG stimulated MYC^{low} cells consumed less amounts of fatty acids than MYC^{high} cells. Furthermore, IL10+CpG stimulated cells are nearly independent in their oxidative phosphorylation from glucose.

Thus, our data suggest that IL10+CpG-stimulated B cells are characterized by glutaminolysis supporting anabolic functions and oxidative phosphorylation is supported in part by both fatty acids and glutamine.

In addition parts of the discussion are changed in that the difference between IL10+CpG stimulated Myc^{low} and Myc^{high} cells can be explained in part by a different need in glutamine for oxydative phosphorylation on page 26 line 552.

... The different glutaminolysis networks identified herein, were IL10+CpG stimulated MYC^{low} cells use less Gln for oxidative phosphorylation compared to MYC^{high} cells fits well with previous data demonstrating that MYC-dependent osteosarcoma cells use glutamine for cellular respiration¹⁰. Therefore, we propose that GOT2 serves as a metabolic hub, providing cells with Asp and/or α -KG for energy demands and biosynthetic processes such as nucleotide biosynthesis in a context-dependent manner. Further, transaminases other than GOT2 such as GPT2 may contribute to the observed differences. GPT2 promotes Gln-dependent anaplerosis by catalyzing the reversible addition of an amino group from glutamate to pyruvate generating Ala and α -KG to fuel the TCA. It is believed that this provides a link between glycolysis, glutaminolysis and TCA anaplerosis. Smith et al. provided evidence for a model in which GPT2 is critical for the cancer phenotype including glutamine-driven TCA anaplerosis and coupling of glycolytic pyruvate output to glutamine catabolism²⁵. Interestingly, GPT2 was dispensable in not fully transformed cells similar as observed herein for IL10+CpG stimulated MYC^{low} cells. Thus it seems to be reasonable that the Warburg effect supports oncogenesis via GPT2-mediated coupling of pyruvate production to glutamine catabolism in MYC^{high} cells but not IL10+CpG stimulated MYC^{low} cells²⁵. It remains to be seen whether the observed oxidative decarboxylation of malate into pyruvate also contributes to the observed differences between IL10+CpG

stimulated MYC^{low} and MYC^{high} cells^{21,49}.

Furthermore, the context-dependence of glutaminolysis was also reflected in the observation that oxidative phosphorylation of IL10+CpG stimulated MYC^{low} cells was nearly independent from transaminase activity and Gln or fatty acids were not as dominantly involved as in MYC^{high} cells. In contrast, MYC^{high} cells are characterized by a strong dependence of oxidative phosphorylation from aminotransferases as observed by AOA treatment but also from fatty acids and glutamine. The lower nitrogen incorporation in Ala in IL10+CpG stimulated MYC^{low} cells as in MYC^{high} cells, a stronger decrease of Ala amounts in MYC^{high} cells after AOA treatment and the effect of the *GPT2* knockdown onto cell proliferation suggest that MYC^{high} much depend much more on anaplerosis to meet their increased energy, biosynthesis and redox needs^{50,51}.

....

3. How are nucleotide levels changed in their stimulations and challenge conditions? These data would support function of GOT2 for nucleotide synthesis (rather than some other function).

Answer: As described above, the recommended alpha-¹⁵N-glutamine labelling analysis revealed no strong differences between IL10+CpG stimulated MYC^{low} and MYC^{high} cells with respect to the measured metabolites.

4. Both Myc and Jak/Stat activation stimulate GOT2 expression, but are these regulatory functions truly independent? Can Jak/stat stimulate metabolism w/o increasing Myc levels? In looking at Myc vs Jak/stat dependence, tracer studies are recommended rather than relying on gene expression, as the functional output is important for demonstrating true regulation.

Answer: This is an absolutely interesting and indeed an important point to discuss. Based on all of our data, not only from this manuscript, we are working on the model that the combined activation of Jak/STAT with NF-κB „substitutes“ Myc as an oncogenic gene expression regulator. To our understanding the ChIP data for GOT2 but also CDK4, SOCS3 or IκBα (Feist et al. 2017) very strongly support the view that both STAT3 and p65 directly regulate their gene expression. We are aware that more sophisticated promoter studies and nascent transcription analysis have to be combined with comprehensive ChIP analysis as performed by the Young, Levens and Amati groups. In addition, in the future it would be interesting to compare directly the occupancy of metabolic gene promoters with Myc vs phosphorylated STAT3/p65. However, the focus of this manuscript is to experimentally model the link between the capacity of microenvironmental factors to activate cell metabolism in support of cell proliferation.

Nevertheless, to answer this question from a different perspective we performed additional analysis. Also answering reviewer 1 we analysed the influence of IL10, CpG and IL10+CpG in

P493-6Myc^{low} cells at different time points at 2, 4, 8 and 24 hours for Myc protein levels. IL10 alone increases Myc protein at very low levels clearly arguing against Jak/STAT-mediated increase in Myc levels. Both CpG alone and IL10+CpG costimulation led to a comparable increase in Myc protein levels with a peak at 4 hours. At time point 24 hours the Myc levels returned to the level of unstimulated cells (**Suppl. FIGURE 9 A**). As the biological outcome of the costimulation is cell proliferation and increased metabolism, the equal small Myc increase by CpG and CpG+IL10 cannot account for this functional difference. More importantly, to test for the influence of this Myc protein increase onto the proliferation of P493-6 cells, we performed a Myc knockdown and measured the cell proliferation (**Suppl. FIGURE 9B,C**). In P493-6Myc^{high} cells the Myc protein amount is reduced about 2 fold, whereas the proliferation is strongly affected. In IL10+CpG stimulated cells the „increase“ in Myc protein is reduced by the knockdown to the level of unstimulated cells, whereas the proliferation is still remarkably high. Relying only on gene expression analysis is indeed insufficient to some extent. Therefore, we analysed the changes in protein amounts and showed that also at the protein level IL10+CpG stimulation lead to increased GOT2 amounts. These data are included in **Suppl. Table 1**. Tracer studies after STAT3/p65 knockdown are surely upcoming investigations for the future. The corresponding data are included into **Figure 6** and the main text was changed in the following way on page 5 line 111:

.... Importantly, protein amounts were also increased (**Suppl. Table 1**). ...

on page 15 line 308:

.... The transaminase GOT2 catalyzes the conversion of α -KG and Asp to Glu and OAA, while GOT1 catalyzes the reverse reaction in the malate-aspartate shuttle as long as the mitochondrial electron transport chain is functional^{20,22,23,24}. As shown in **Suppl. Table 1** the abundance of both enzymes GOT1 and GOT2 is increased after IL10+CpG stimulation of MYC^{low} cells. Thus, the transcriptional activation is also translated into protein. ...

The changes about our analysis on the role of IL10+CpG „activated“ Myc are included in the main text as follows on page 20 line 417:

..... To exclude that increased cell proliferation of IL10+CpG stimulated MYC^{low} cells was rather the result of increased MYC expression mediated by STAT3 and p65 activation, both the changes in Myc protein abundance as well as the effect of RNAi mediated Myc knockdown onto cell proliferation were monitored over 24 hours (**Suppl. Figure 9**). A transient low activation of Myc was observed after both CpG and IL10+CpG stimulation of MYC^{low} cells. However, cell proliferation was only increased upon IL10+CpG costimulation (**Suppl. Figure 9A**). Furthermore, the knockdown of Myc affected the proliferation of MYC^{high} cells much stronger despite a less pronounced reduction in Myc (**Suppl. Figure 9B, C**). Considering STAT3/p65 ChIP analysis, Jak/STAT and NF- κ B inhibition and Myc knockdown data, we conclude that the synergistic effect of IL10 and CpG is driven mainly by STAT3 and NF- κ B and not by Myc..

Performing additional pathway analysis, we observed an further level of bringing together both Jak/STAT and NF-κB pathways activated by IL10+CpG:

- IL10 activation mediates Tyr-phosphorylation of STAT3, whereas
- in addition to NF-κB TLR9 activation leads to the Ser-Phosphorylation of STAT3 in MYC^{low} cells.

Thus, it looks like that not only the combined binding of STAT3/p65 to gene promoters explains the pathway synergy but also the specific posttranslational modification of STAT3 by both pathways. Based on literature this is leading to more active STAT3 and it can be suggested as an important prerequisite for the interaction with p65 on corresponding gene promoters.

Therefore, within the main text the following paragraph was included on page 20 line 412:

... In addition, analysis of STAT3 phosphorylation in MYC^{low} cells showed, that IL10 and CpG mediated the respective phosphorylation of Tyr-705 and Ser-727 (**Figure 6E**). Hence, both coordinated STAT3 phosphorylation, which is a known prerequisite for the interaction of STAT3 with p65, and the joint binding of STAT3 and p65 NF-κB to their target genes mechanistically support the synergy of IL10 and CpG (**Figure 6E**)²⁶.

And on page 25 line 532 within the discussion:

.... In addition, IL10 and CpG modulate STAT3 activity via phosphorylation of amino acid residues Tyr-705 and Ser-727 in STAT3. Therefore, it is reasonable to conclude that both the coordinated STAT3 phosphorylation and joint binding of STAT3 and p65 NF-κB to their target genes are important in the observed pathway synergy of both microenvironmental factors. Future studies may reveal additional posttranslational modifications in both STAT3 and p65²⁶. ..

5. The data in Fig 4B is informative, but I am concerned about the low level of knockdown. Is Got1 really not important for Asp synthesis and proliferation here? With 30-40% expression remaining, there may not be a tangible effect on flux.

Answer: P493-6 cells are difficult to transfect. With siRNA it is a bit easier compared to plasmid DNA. Compared to STAT3 all other so far used siRNAs are less efficient in reducing the amount of corresponding target proteins. The effect of the GOT knockdowns onto cell proliferation is indeed lower as expected. Currently, it is difficult to estimate what is the lowest level of protein necessary to maintain metabolite flux in these cells. From the proteomics analysis shown in Suppl. Table 1 we know that the amount of both enzymes is increased by IL10&CpG or Myc. The role of GOT1 in IL10+CpG stimulated cells is indeed puzzling and whether remaining 30% of GOT1 are sufficient to support metabolic fluxes is unclear but difficult to test in this cell system. As GOT1 expression is not different between normal B cell subsets and lymphomas in contrast to GOT2 we decided to focus our analysis onto GOT2.

In addition, the relative low effect of the GOT knockdowns onto cell proliferation in comparison to

AOA treatment or even knockdown of STAT3 and p65 suggests that both transaminases are involved in the metabolic support of cell proliferation but other transaminases might be necessary to interrupt the glutaminolysis in support of proliferation markedly. Therefore, we performed additional knockdowns of for example mitochondrial glutamic pyruvate transaminase 2 (GPT2) and included these data into **Figure 4**.

The main text is changed in the following way on pages 15 line 326:

.... Differences between the effects of AOA treatment and GOT2 knockdown indicated the involvement of other transaminases. This prompted us to investigate the glutamate pyruvate transaminase (GPT2), the gene expression of which had increased less in IL10+CpG stimulated MYC^{low} cells as in MYC^{high} cells. No significant decrease in cell proliferation was observed after GPT2 knockdown in IL10+CpG stimulated MYC^{low} cells. In contrast cell proliferation of MYC^{high} cells was decreased after GPT2 knockdown (**Figure 4E/F**). A specific role for GPT2 in MYC^{high} cells is in line with the above described higher nitrogen incorporation into Ala and the decrease in Ala abundance after AOA treatment and the recent description of cells with cooperating oncogenic mutations ²⁵....

and within the discussion on page 26 line 552 and page 28 line 604:

... The different glutaminolysis networks identified herein, were IL10+CpG stimulated MYC^{low} cells use less Gln for oxidative phosphorylation compared to MYC^{high} cells fits well with previous data demonstrating that MYC-dependent osteosarcoma cells use glutamine for cellular respiration ¹⁰. Therefore, we propose that GOT2 serves as a metabolic hub, providing cells with Asp and/or α -KG for energy demands and biosynthetic processes such as nucleotide biosynthesis in a context-dependent manner. Further, transaminases other than GOT2 such as GPT2 may contribute to the observed differences. GPT2 promotes Gln-dependent anaplerosis by catalyzing the reversible addition of an amino group from glutamate to pyruvate generating Ala and α -KG to fuel the TCA. It is believed that this provides a link between glycolysis, glutaminolysis and TCA anaplerosis. Smith et al. provided evidence for a model in which GPT2 is critical for the cancer phenotype including glutamine-driven TCA anaplerosis and coupling of glycolytic pyruvate output to glutamine catabolism ²⁵. Interestingly, GPT2 was dispensable in not fully transformed cells similar as observed herein for IL10+CpG stimulated MYC^{low} cells. Thus it seems to be reasonable that the Warburg effect supports oncogenesis via GPT2-mediated coupling of pyruvate production to glutamine catabolism in MYC^{high} cells but not IL10+CpG stimulated MYC^{low} cells ²⁵. It remains to be seen whether the observed oxidative decarboxylation of malate into pyruvate also contributes to the observed differences between IL10+CpG stimulated MYC^{low} and MYC^{high} cells ^{21,49}.

... Aminotransferases facilitate amino acid-mediated TCA anaplerosis and are emerging as critical determinants of oncogenesis. Recently GOT1, GPT2, the branched-chain amino acid transferase and phosphoserine aminotransferase have been found to be essential for tumourigenesis of pancreatic adenocarcinoma, glioblastoma, and breast cancer cells, respectively further supporting the herein described context dependent role of GOT2 but also GOT1 and GPT2 ^{22,25,60,61}. ...

Minor

The authors should read (and cite) the manuscript by Murphy et al. (Met Eng 2013), which like the pub by Le et al. performed detailed ¹³C tracer analysis on the cell lines used here.

Answer: We aware about the work of Murphy et al. and discussed it also before. Both Murphy et al. and Le et al. are focused onto Myc, whereas we are more interested in the synergy of IL10+CpG. Based on additional analysis and your and reviewer 1 suggestions we cited Murphy et al. as for example within the main results part and the discussion. On page 5 line 115 and 26 line 568:

.... Next, we measured both uptake of Gln and glucose (Glc) and secretion of lactate to determine the lactate-to-glucose molar ratio (L/G) (**Suppl. Figure 3**). Strikingly, the uptake of both Gln and Glc was significantly increased in IL10+CpG-stimulated MYC^{low} cells compared to cells exposed to only one stimulus or none at all and even more so in MYC^{high} cells (**Suppl. Figure 3A, B**). The L/G molar ratio was similar across the analysed conditions and indicated that glucose is predominantly converted to lactate as reported before (**Suppl. Figure 3C**)²¹. However, it has to be taken into account that both glucose consumption and lactate production are lower in IL10+CpG-stimulated MYC^{low} cells compared to MYC^{high} cells.

... It remains to be seen whether the observed oxidative decarboxylation of malate into pyruvate also contributes to the observed differences between IL10+CpG stimulated MYC^{low} and MYC^{high} cells^{21,49}

Line 202: why is thymine crossed out?

Answer: We apologize for this formatting error and corrected it.

REVIEWERS' COMMENTS:

Reviewer #1 (Remarks to the Author):

The authors have substantively addressed my concerns.

Reviewer #2 (Remarks to the Author):

The authors have revised their MS and responded to specific questions I raised in the review process.

- 1) I asked about GOT activity as opposed to gene expression. The authors have added GOT protein levels, this does not equal GOT activity. This is easily measured and seems critical in a MS claiming a critical role for the transcriptional regulation of this enzyme as an oncogene and target.
- 2) Thanks for clarifying the MYC environment.
- 3) GOT expression in various tissues. The authors have added data showing GOT2 expression in lymphomas, the question is really where else is GOT2 expressed? Is this a liver, cardiac etc enzyme. This is relevant to the expected toxicity. Should be easy to stain some mouse organs or human tissue sections.
- 4) Title will need editorial advice.
- 5) I asked about synergy versus additivity. There are simple statistical tools to distinguish the two. The authors should just use them.
- 6) I asked about the reported role of uptake into ornithine and the results seem to be removed.
- 7) Additional cell lines have been added.
- 8) My question about the compounds is answered.
- 9) The therapeutic aspects of the study remain unconvincing.

Overall, many points have been addressed, others remain. A key issue for me is GOT2 activity. I don't think the therapeutic implications are sufficiently supported by data.

Reviewer #3 (Remarks to the Author):

The authors have addressed my critiques.

RESPONSE TO REVIEWERS' COMMENTS:

Reviewer #1 (Remarks to the Author):

The authors have substantively addressed my concerns.

Reviewer #2 (Remarks to the Author):

The authors have revised their MS and responded to specific questions I raised in the review process.

1) I asked about GOT activity as opposed to gene expression. The authors have added GOT protein levels, this does not equal GOT activity. This is easily measured and seems critical in a MS claiming a critical role for the transcriptional regulation of this enzyme as an oncogene and target.

Answer: We performed additional experiments to measure the activity of GOT2 according to the colorimetric protocol described by Frankel and Reitman (1957) and by ¹H NMR spectroscopy of aspartate and alpha-ketoglutarate. These data are included in Supplementary Figure 9.

The main text has been modified as follows (page 11):

"The abundance and catalytic activity of GOTs is increased after IL10+CpG stimulation of MYC_{low} cells as shown in Supplementary Table 1 and Supplementary Fig. 9. Thus the transcriptional activation (Figure 1c) is also translated into protein and enzymatic activity."

2) GOT expression in various tissues. The authors have added data showing GOT2 expression in lymphomas, the question is really where else is GOT2 expressed? Is this a liver, cardiac etc enzyme. This is relevant to the expected toxicity. Should be easy to stain some mouse organs or human tissue sections.

Answer: Within Figure 7B-D it is visible that GOT2 is specifically expressed in lymphoma cells. Within tonsils the T cell zone is also mainly GOT2 negative. Within the B cell follicle not all cells are showing positive GOT2 stainings. Additionally, we performed immunohistochemical staining of liver, lung, muscle and skin specimens with anti-GOT2 antibody. As expected, liver tissue exhibits strong GOT2 expression. Within the basal layer of the skin, nuclear staining of most likely nonspecific nature is observed. Data have been included in Supplementary Figure 12. The corresponding figure legend reads as follows:

"Supplementary Figure 12: Immunohistological staining of GOT2 in representative specimens of (a) liver, (b) muscle, (c) skin, and (d) lung. The bar in (a) is 50 μ m. Please note the homogeneous GOT2 staining in the cytosol of hepatocytes. The nuclear staining in skin cells seems to be nonspecific."

The legend to Figure 7 was extended as follows: " The scale bar represents 50 μ m. In Supplemental Fig. 12 a corresponding GOT2 staining for liver, lung, muscles and skin is provided showing a homogeneous GOT2 expression in hepatocytes. "

3) Title will need editorial advice.

Answer: The title was changed as suggested: "Cooperative STAT/NF- κ B signaling regulates lymphoma metabolic reprogramming and aberrant GOT2 expression."

4) I asked about synergy versus additivity. There are simple statistical tools to distinguish the two. The authors should just use them.

Answer: We now indicate in Figure 1 if the interaction term of IL10 and CpG was significant in the individual gene models. The corresponding figure legend reads as follows:
„Effects on gene expression were analyzed by linear regression and p-values for the IL10:CpG interaction terms were calculated (adjusted by Benjamini-Hochberg). Positive synergistic interactions are marked with a star (* $p < 0.05$).“

For Suppl. Figure 1, we used a regularized regression approach (lasso) because the high number of coefficients in these models quickly leads to unpredictable models when using regression without regularization. The models we show have high predictive power ($R^2 > 0.8$) and therefore, all coefficients in the model constitute important components needed to predict gene expression. We believe that predictive performance indicates significance better than a p-value could. Moreover, the interaction term IL10:CpG appears in 81% of the models shown, indicating its global importance in explaining gene expression variation induced by the stimuli. We added this analysis to Supplementary Figure 1: „For 81% of the gene models, the Lasso selected the term of the interaction between IL10 and CpG, indicating its global importance when explaining gene expression changes by the environmental stimuli.“

5)The therapeutic aspects of the study remain unconvincing.

Answer: We have de-emphasized the potential therapeutic aspects of our study. Based on the concerns of this reviewer we rewrote the corresponding part of the abstract in the following way: “ Furthermore, high aberrant GOT2 expression is prognostic in diffuse large B-cell lymphoma underscoring the current findings and importance of stromal factors in lymphoma biology.”

6)Overall, many points have been addressed, others remain. A key issue for me is GOT2 activity. I don't think the therapeutic implications are sufficiently supported by data.

Answer: See above.

Reviewer #3 (Remarks to the Author):

The authors have addressed my critiques.